# Ongoing declines for the world's amphibians in the face of emerging threats

Systematic assessments of species extinction risk at regular intervals are necessary for informing conservation action[1,2]. Ongoing developments in taxonomy, threatening processes and research further underscore the need for reassessment[3,4]. Here we report the findings of the second Global Amphibian Assessment, evaluating 8,011 species for the International Union for Conservation of Nature Red List of Threatened Species. We find that amphibians are the most threatened vertebrate class (40.7% of species are globally threatened). The updated Red List Index shows that the status of amphibians is deteriorating globally, particularly for salamanders and in the Neotropics. Disease and habitat loss drove 91% of status deteriorations between 1980 and 2004. Ongoing and projected climate change effects are now of increasing concern, driving 39% of status deteriorations since 2004, followed by habitat loss (37%). Although signs of species recoveries incentivize immediate conservation action, scaled-up investment is urgently needed to reverse the current trends.

The International Union for Conservation of Nature (IUCN) Red List Index (RLI) documents the extinction risk trends of species groups over time[5], generating information that is crucial for conservation prioritization and planning[6]. The landmark 2004 Global Amphibian Assessment (GAA1) was published on the IUCN Red List, demonstrating that amphibians were the most threatened class of vertebrates worldwide, and has been widely used to guide and motivate amphibian conservation efforts[7]. The 2004 baseline study identified habitat loss and degradation and over-exploitation as the main threats, contributing to the deterioration of just over half of the species that deteriorated in status between 1980–2004, while 48% were classified as enigmatic-decline species[7]. Subsequent studies support that the disease chytridiomycosis, caused by *Batrachochytrium dendrobatidis*, was most likely responsible for many enigmatic declines[8–12]. The GAA1 helped to launch a wave of research and conservation efforts directed at *B. dendrobatidis* and the other threats causing the decline in amphibians[6].

Completed in June 2022, the second Global Amphibian Assessment (GAA2) reassessed the status of the GAA1 species and added 2,286 species, bringing the number of amphibians on the IUCN Red List to 8,011 (39.9% increase from 2004; covering 92.9% of 8,615 described species). Since the GAA1, information on population trends, ecological requirements, threats and distributional boundaries of amphibians has improved considerably, and amphibian systematics have progressed. However, this new information (for example, better estimates of population size, redefining taxonomic boundaries) can sometimes result in a non-genuine change in Red List category, introducing biases in the data. We therefore used current information to estimate a backcasted Red List category for each species in 1980 and 2004 and examine only genuine category changes. With these data and the GAA2 assessments, we re-examine the global status and trends of amphibians and present new insights on threats, providing a crucial update that informs the prioritization, planning and monitoring of conservation actions.

## Threatened and extinct species

The status of amphibians worldwide continues to deteriorate: 40.7% (2,873) are globally threatened (that is, IUCN Red List categories Critically Endangered, Endangered and Vulnerable), compared with 37.9% (2,681) in 1980 and 39.4% (2,788) in 2004 (Fig. 1 and Extended Data Table 1; see the 'Percentage of threatened species' section of the Methods). The proportion of species in the Data Deficient IUCN category has decreased from 22.5% in the GAA1 to 11.3% as a result of newly available information.

The greatest concentrations of threatened species are in the Caribbean islands, Mesoamerica, the Tropical Andes, the mountains and forests of western Cameroon and eastern Nigeria, Madagascar, the Western Ghats and Sri Lanka. Other notable concentrations of threatened species occur in the Atlantic Forest biome of southern Brazil, the Eastern Arc Mountains of Tanzania, central and southern China, and the southern Annamite Mountains of Vietnam (Fig. 1). Of all of the comprehensively assessed groups on the IUCN Red List, amphibians are the second most threatened group and remain the most threatened vertebrate class (cycads, 69%; sharks and rays, 37.4%; conifers, 34.0%; reef-building corals, 33.4%; mammals, 26.5%; reptiles, 21.4%; dragonflies, 16%; birds, 12.9%; cone snails, 6.5%)[13–19].

Documented amphibian extinctions continue to increase: there were 23 by 1980, an additional 10 by 2004 and four more by 2022, for a total of 37 (Extended Data Table 1). The most recent are *Atelopus chiriquiensis* and *Taudactylus acutirostris*, after rapid declines linked to chytridiomycosis in the 1990s, while *Craugastor myllomyllon* and *Pseudoeurycea exspectata* were last seen in the 1970s and are believed to be Extinct due to agricultural expansion. Strict requirements must be met to declare a species Extinct[20]; therefore, many species missing for decades are categorized as Critically Endangered (CR) and tagged as Possibly Extinct (CR(PE)). For 1980, 24 amphibians were categorized as CR(PE), for 2004 this increased to 162, with another 23 added for 2022 (Extended Data Table 1). Thus, the number of known amphibian

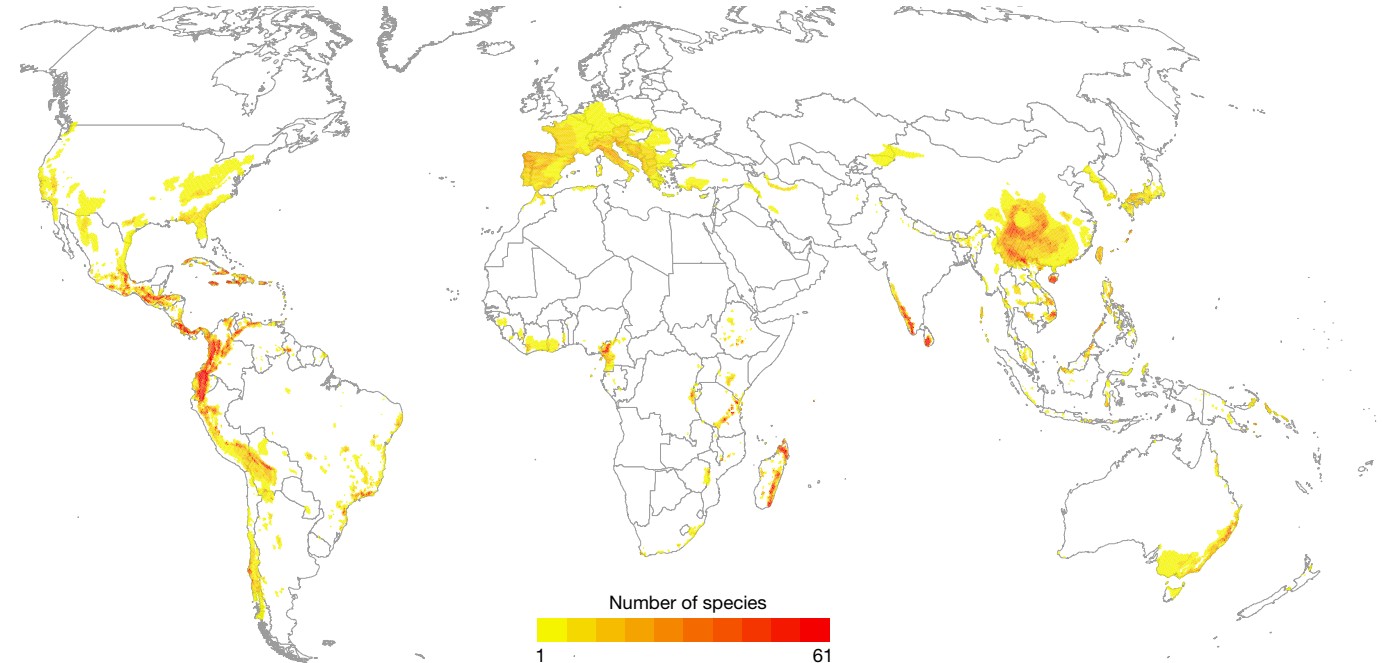

**Fig. 1 | The distribution of 2,873 globally threatened amphibian species.** The darker colours correspond to higher species richness. The colour scale is based on 10 quantile classes. Maximum richness equals 61 species. The cell area is 865 km². One species was excluded because no spatial data were available.

extinctions could be as many as 222 over the last 150 years if all CR(PE) species are indeed extinct.

When considering all threatened amphibians, the most commonly documented threats are types of habitat loss and degradation, with the top three being agriculture (77% of species impacted), timber and plant harvesting (53%), and infrastructure development (40%) (Fig. 2). Climate change effects (29%) and disease (29%) are other common

threat types. Although these are important findings, they do not account for the severity and scope of these threats.

## The RLI

The RLI is an indicator calculated from Red List categories to measure trends in extinction risk over time[5]. RLI values range from 1 (all species

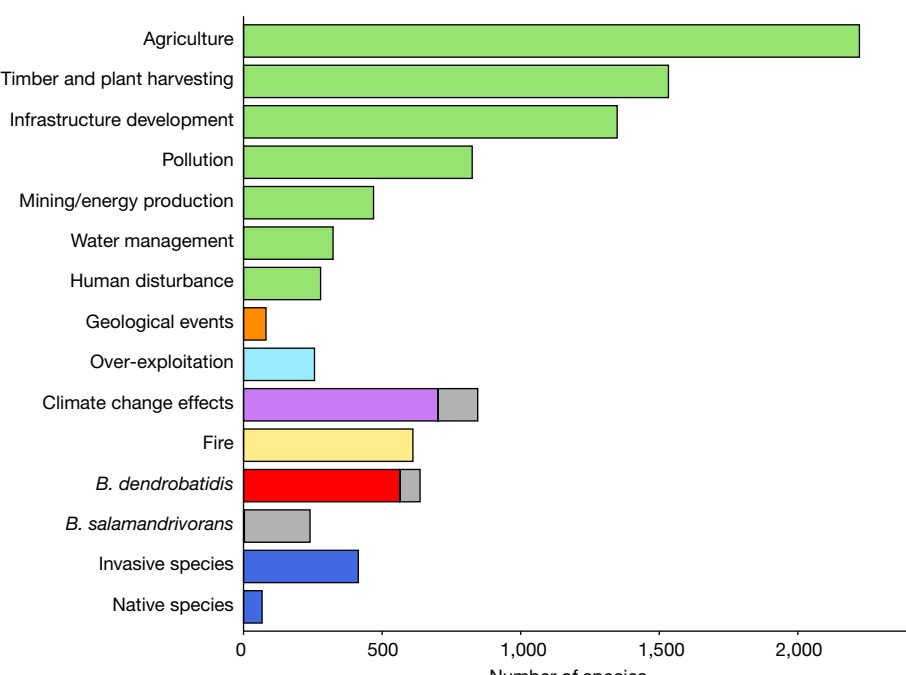

**Fig. 2 | The types of threats affecting amphibian species in threatened categories.** The threats to a species were coded using the threat-classification scheme and grouped for ease of comparison (see the 'Classification schemes' and 'Threats to threatened species' sections of the Methods). All threats

shaded in green are causing habitat loss and degradation. The grey sections denote the number of species for which the threat timing is in the future rather than ongoing. Note that most species are experiencing multiple threats.

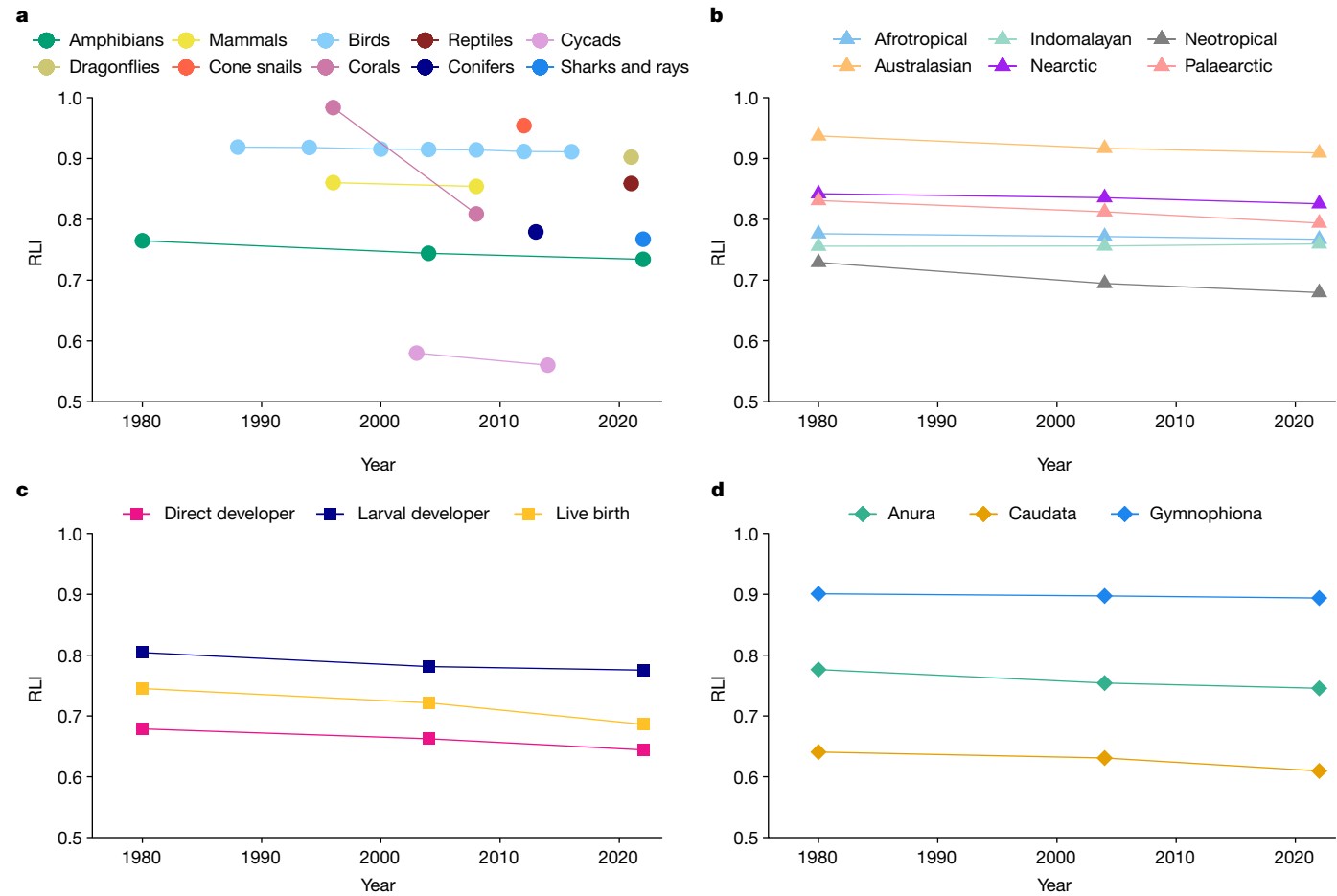

**Fig. 3 | RLIs showing trends in overall extinction risk. a**, The RLIs of all comprehensively assessed taxonomic groups on the IUCN Red List. **b**, The amphibian RLI disaggregated by biogeographical realm. **c**, The amphibian RLI disaggregated by breeding strategy. **d**, The amphibian RLI disaggregated by order.

are Least Concern) to 0 (all are Extinct). A change in the value is influenced only by species moving between categories due to genuine improvements or deteriorations in status, with non-genuine category changes excluded through backcasting (see the 'RLI' section of the Methods). The RLI was calculated for amphibians for 1980, 2004 and 2022 using the data collected in this study, and compared to other species groups[13] (Fig. 3a). A negative RLI trend is observed in all groups with more than one RLI datapoint, indicating that the number of species in higher extinction risk categories is increasing (Fig. 3a). Although the amphibian RLI trend between 2004 and 2022 is slightly less steep compared with the previous period, it continues to decline.

Trends in extinction risk differ across biogeographical realms (Fig. 3b and Extended Data Table 3). The Neotropics (with 48% of amphibians) has the lowest RLI value of all realms and has the greatest deterioration in status, although the gradient lessens during 2004–2022. The Neotropical trend is associated with chytridiomycosis outbreaks in the 1970s–2000s, with many of the most susceptible species affected before 2004. Australasia has the highest RLI, primarily because there are comparatively fewer threats to the large number of species on New Guinea, which is currently a chytridiomycosis-free refuge[21] with a reasonable possibility of a period of outbreak and decline in the future. The Palaearctic and Nearctic RLIs show accelerating declines during 2004–2022. In the Palaearctic, habitat loss and degradation is the leading cause followed by the emerging threat of the fungal pathogen *Batrachochytrium salamandrivorans*, whereas, in the Nearctic, climate change effects are the most common cause, followed by habitat loss and degradation. The RLI trend for the Afrotropics is declining

across both periods, initially driven by habitat loss/degradation but, more recently, disease emerges as the most common cause. The Indomalayan RLI trend shows a slight improvement between 2004 and 2022, probably due to the creation and improved management of protected areas.

Among the three most common breeding strategies for amphibians, extinction risk is higher for direct developers than for larval developers and live bearers (Fig. 3c and Extended Data Table 3; see the 'Breeding strategy' section of the Methods). The RLI of all three groups declined at a similar rate between 1980 and 2004. However, during 2004–2022, it slows for larval developers and slightly accelerates for live bearers and direct developers. This result is probably due to larval developers having been especially impacted by *B. dendrobatidis* before 2004 when chytridiomycosis outbreaks were at their peak (particularly in high-elevation streams). The causes of differing extinction risks between breeding strategies merit further study.

Extinction risk also exhibits important phylogenetic patterns (Fig. 3d and Extended Data Table 3). The RLI for Caudata (salamanders and newts) is consistently the lowest, making them the most threatened. Although the RLI for Caudata declined at a lesser rate than for Anura (frogs) during 1980–2004, the rate of decline increased between 2004–2022. By contrast, the RLI for Anura declined at a much greater rate between 1980 and 2004, but at a lesser rate between 2004 and 2022, probably due to the timing of global chytridiomycosis outbreaks. A slight downward trend is shown for Gymnophiona (caecilians) with the caveat that they are very poorly studied: only 115 out of the 206 assessed are included in the RLI due to 44% being categorized as data deficient and 17% are threatened.

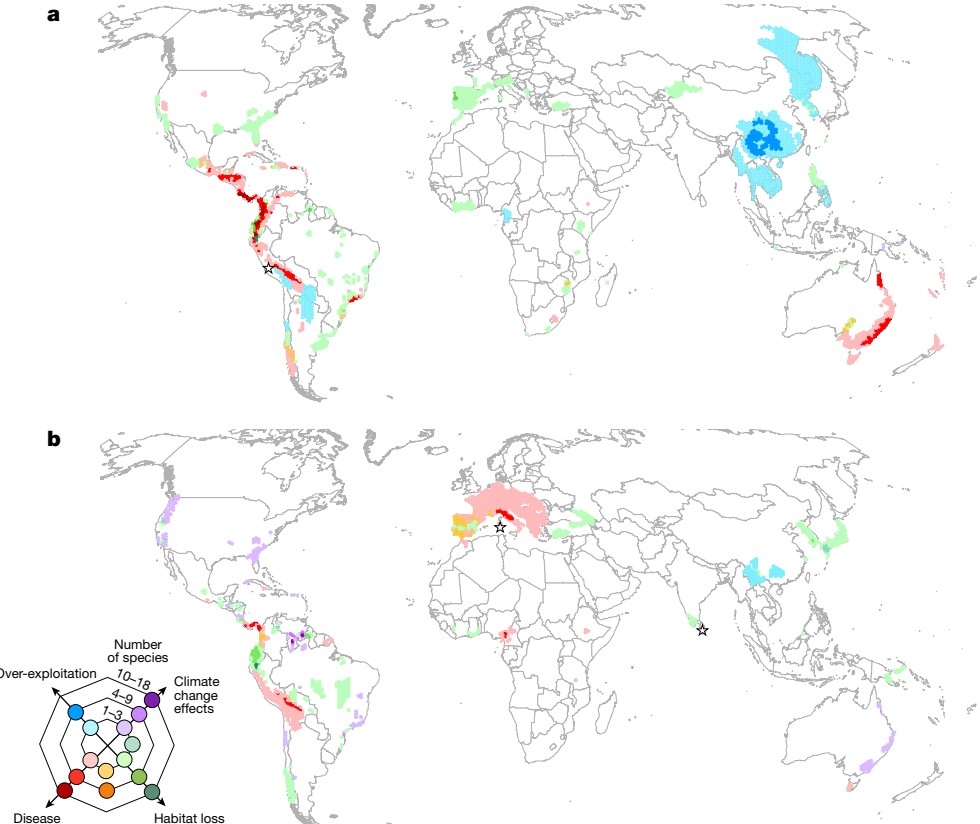

**Fig. 4 | Geographical pattern of the primary drivers of deteriorating status among amphibians. a,b,** The primary drivers of deteriorating status among amphibians during 1980–2004 (482 species; **a**) and 2004–2022 (306 species; **b**). Cell colour was determined by the primary driver impacting the most species.

Where two primary drivers equally contribute to a cell, an intermediate colour is shown. The stars indicate where the primary driver is undetermined or there are numerous primary drivers. The cell area is 7,775 km².

## Genuine changes in status

To better understand which threats are driving deteriorations in status, the subset of species that changed Red List categories over time were examined further. For each species in the subset, the threat that contributed most substantially to the deterioration in status was determined and defined as the primary driver. These are categorized into four main groups: disease, climate change effects, habitat loss/degradation and over-exploitation (Extended Data Table 2; see the 'Grouping of primary drivers' section of the Methods). Since 1980, 87% of category changes involved a change into a higher extinction risk category, with 482 of those changes occurring between 1980 and 2004 (Supplementary Table 3a) and 306 between 2004 and 2022 (Supplementary Table 3b).

The geographical pattern of primary drivers for amphibians with a deteriorating status is not uniform (Fig. 4). Disease was the primary driver for 281 species (58%) during 1980–2004, compared with 69 species (23%) during 2004–2022 (Extended Data Table 2). Disease is recorded as the dominant primary driver of status deteriorations from Costa Rica to the Andes of South America during 1980–2004 and 2004–2022, while newer hotspots of disease-related declines are appearing in central and eastern Africa (Fig. 4). *B. salamandrivorans* is an emerging threat in Europe (Fig. 4b), where status deteriorations are being driven by projected declines for some species.

There are some interesting points of difference when comparing the current distribution map of all threatened species (Fig. 1) to the distribution of species that have deteriorated in status between 2004 and 2022 (Fig. 4b). Several global hotspots for threatened amphibians such as Madagascar, Hispaniola, the Eastern Arc Mountains of Tanzania

and the southern Annamite Mountains of Vietnam are notably absent from the map of species that deteriorated in status. In these regions, threats have been ongoing for decades, and many species are already considered to be highly threatened. For example, deteriorations in status due to disease and high rates of habitat loss on Hispaniola are apparent in the previous time period 1980–2004 (Fig. 4a), with a large proportion of species endemic to the island already on the brink of extinction at the time GAA1 was completed. On the contrary, other global hotspots for threatened amphibians continue to experience status deteriorations. Two of the most speciose regions of the world for amphibians—the Tropical Andes and Mesoamerica—have held considerable numbers of species that have deteriorated in status since 1980.

Species moving into the highest extinction risk categories are much more likely to have been affected by disease (Fig. 5), as chytridiomycosis results in rapid and widespread population declines for susceptible species[9,10]. Disease is the primary driver for 76% of category changes to CR and 79% of changes to CR(PE) between 1980–2004 and remains the primary driver pushing species into CR(PE) between 2004 and 2022 (89%; Fig. 5). By contrast, status deteriorations due to projected climate change effects are more frequently into categories of lower extinction risk (that is, Near Threatened or Vulnerable).

Climate change effects are the most common primary driver of status deteriorations during 2004–2022, with 119 species (39%) affected compared with 6 species (1%) during 1980–2004 (Fig. 4 and Extended Data Table 2). A notable example is the amphibians endemic to Venezuelan tepuis (table-top mountains) (Fig. 4b and Supplementary Table 1), which are particularly vulnerable to predicted habitat shifting due to climate change because vertical migration and dispersal are

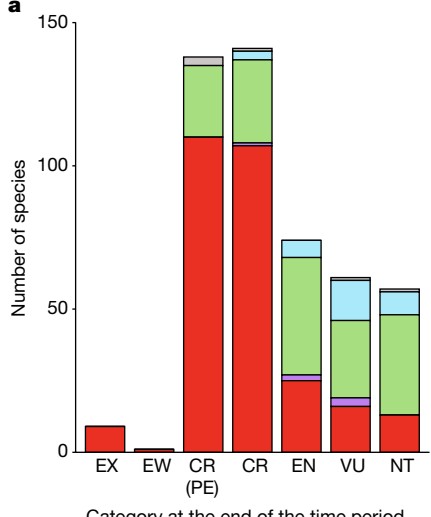

**Fig. 5 | Species moved into a higher Red List category coded by the primary driver causing the change. a,b,** The number of species moved into a higher Red List category, coded by the primary driver causing the change, during 1980–2004 (**a**) and 2004–2022 (**b**). Red List categories are ordered by highest to lowest threat level: Extinct (EX), Extinct in the Wild (EW), Critically Endangered (CR), Endangered (EN), Vulnerable (VU) and Near Threatened (NT). CR species that are likely to be extinct have the Possibly Extinct (PE) tag.

impossible. Decreased rainfall due to climate change in the Wet Tropics of Australia and Brazil's Atlantic Forest is also predicted to reduce the reproductive success of direct-developing frogs (for example, in the genera *Cophixalus* and *Brachycephalus*) owing to their dependence on high levels of soil and leaf-litter moisture to prevent egg desiccation. In eastern Australia and western United States, climate change is increasing the frequency, duration and severity of droughts and fires[22], often compounding existing threats from disease and habitat loss. For example, five US salamander species in the genus *Batrachoseps* have deteriorated in status due to the increasing effects of fires and reduced soil humidity. Given the scarcity and geographical bias of studies on the effects of climate change on amphibians[23], the true impacts are probably underestimated. As further studies are published and climate change effects continue to increase and intensify, the status of additional amphibians is expected to deteriorate.

Habitat loss and degradation remains the most prevalent primary driver of status deteriorations in many regions (156 species or 32% in 1980–2004, 112 species or 37% in 2004–2022) (Extended Data Table 2). Between 2004 and 2022, hotspots caused by ongoing or projected habitat loss are prominent in the Andes of Ecuador, central Guyana and Republic of Korea (Fig. 4b).

Although most category changes since 1980 are deteriorations (788), 120 species have shown improvements in status, moving to less-threatened Red List categories (Extended Data Fig. 1 and Supplementary Table 4a,b). Conservation actions are responsible for 63 of these improvements, 94% of which are results from effective habitat protection and improved habitat management in regions such as the Western Ghats in India, Costa Rica and Sabah in Malaysia.

Another 57 species (largely from the Neotropics and Australia) improved unaided, most of which are now persisting and, in some cases, recovering after experiencing a rapid decline associated with chytridiomycosis. It is evident that there are still no definitive conservation measures known to prevent ongoing decline from disease in wild populations, although many of these species can benefit from habitat protection. For example, some species that previously experienced declines due to disease, but are now persisting, have improved in status because their habitat has remained protected (for example, the Australian species *Litoria aurea*, *Litoria dayi*, *Litoria nannotis*, *Litoria pearsoniana*, *Litoria raniformis* and *Litoria rheocola*). Whereas other species that are persisting after *B. dendrobatidis*-associated declines

may not experience an improvement in category if high rates of habitat loss and degradation are present within their distributions.

## Discussion

The findings of this study confirm that the global amphibian extinction crisis has not abated. Crucially, the primary driver of status deteriorations is shifting from disease to the emerging threat of climate change. This is of particular concern because it often exacerbates other threats, such as land-use change, fire or disease[24–26]. Thus, the GAA2 results highlight the need to investigate and implement conservation actions that address the species-specific effects of climate change, particularly for species identified as imminently at risk of serious population declines.

This study also reinforces that effective habitat protection continues to be a priority for amphibian conservation, as it contributed to the greatest number of status improvements since 1980. However, more amphibians are threatened with extinction than ever before, underscoring the urgency of halting the destruction and degradation of their habitats. Critically, the legal and illegal expansion of agriculture, including animal agriculture and cash crops, is the single most important threat to amphibians worldwide (Fig. 2). The effective protection of globally important sites for amphibians, including Alliance for Zero Extinction sites and other Key Biodiversity Areas[27] (two conservation tools that draw on IUCN Red List data), can safeguard remaining habitat for threatened or geographically restricted species.

The GAA2 data also demonstrate that effective habitat protection alone is not always sufficient in addressing the threats of disease, over-exploitation or climate change effects, as many threatened amphibians already occur within protected areas. Thus, the integration of priority amphibian sites within the wider landscape, to ensure connectivity and enable dispersal, will be important in the face of global change scenarios, as has also been suggested by other studies[28,29]. Furthermore, to avoid a second global amphibian pandemic, which has the potential to trigger a new wave of status deteriorations similar to those related to *B. dendrobatidis* (Figs. 4a and 5a), preventing the spread of *B. salamandrivorans* throughout Europe and its introduction into the Americas is essential[30–32]. Monitoring populations for other new disease risks[33] and developing practical disease management tools are also recommended. Integrating ex situ measures into conservation

plans can buy time[34], especially for the 798 CR species that are at the highest risk of extinction.

The large proportion of Data Deficient amphibians (909 species) continues to require further research to determine their extinction risk and conservation needs (see the 'Data Deficient species' section of the Methods). Many of these are likely to be threatened[35–37]. More broadly, increased population monitoring worldwide[38] is crucial to informing conservation actions and future reassessments. These with other recommended actions are highlighted in the IUCN SSC Amphibian Conservation Action Plan[39].

In support of the conservation actions above, policy responses to the ongoing amphibian extinction crisis, and the biodiversity crisis as a whole, need to be strengthened. Increased political will and sufficient resource commitments for the delivery of agreed global and national biodiversity conservation targets are necessary for the future survival and recovery of this amazingly diverse group of animals.

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

Jennifer A. Luedtke[1,2 ✉], Janice Chanson[1,2], Kelsey Neam[1,2], Louise Hobin[2], Adriano O. Maciel[3], Alessandro Catenazzi[4,5], Amaël Borzée[2,6], Amir Hamidy[7], Anchalee Aowphol[8], Anderson Jean[9,10], Ángel Sosa-Bartuano[11], Ansel Fong G.[12], Anslem de Silva[13], Antoine Fouquet[14], Ariadne Angulo[2], Artem A. Kidov[15], Arturo Muñoz Saravia[16,17], Arvin C. Diesmos[18,19], Atsushi Tominaga[20,21], Biraj Shrestha[22,23], Brian Gratwicke[24], Burhan Tjaturadi[25], Carlos C. Martínez Rivera[26,27], Carlos R. Vásquez Almazán[28,29], Celsa Señaris[30], S. R. Chandramouli[31], Christine Strüssmann[32], Claudia Fabiola Cortez Fernández[16,33], Claudio Azat[34], Conrad J. Hoskin[35], Craig Hilton-Taylor[36], Damion L. Whyte[37], David J. Gower[38], Deanna H. Olson[39], Diego F. Cisneros-Heredia[40,41], Diego José Santana[42], Elizah Nagombi[43], Elnaz Najafi-Majd[44], Evan S. H. Quah[45,46], Federico Bolaños[47,48], Feng Xie[49], Francisco Brusquetti[50], Francisco S. Álvarez[51], Franco Andreone[52], Frank Glaw[53], Franklin Enrique Castañeda[54], Fred Kraus[55], Gabriela Parra-Olea[56], Gerardo Chaves[48], Guido F. Medina-Rangel[57], Gustavo González-Durán[58], H. Mauricio Ortega-Andrade[59,60], Iberê F. Machado[61], Indraneil Das[62], Iuri Ribeiro Dias[63], J. Nicolas Urbina-Cardona[64], Jelka Crnobrnja-Isailović[65], Jian-Huan Yang[66], Jiang Jianping[49], Jigme Tshelthrim Wangyal[67,68], Jodi J. L. Rowley[69,70], John Measey[71,72], Karthikeyan Vasudevan[73], Kin Onn Chan[46], Kotambylu Vasudeva Gururaja[74], Kristiina Ovaska[75,76], Lauren C. Warr[77], Luis Canseco-Márquez[78], Luís Felipe Toledo[79], Luis M. Díaz[80], M. Monirul H. Khan[81], Madhava Meegaskumbura[82], Manuel E. Acevedo[83], Marcelo Felgueiras Napoli[84], Marcos A. Ponce[85], Marcos Vaira[86], Margarita Lampo[87,88], Mario H. Yánez-Muñoz[89], Mark D. Scherz[90], Mark-Oliver Rödel[91], Masafumi Matsui[92], Maxon Fildor[9], Mirza D. Kusrini[93], Mohammad Firoz Ahmed[94], Muhammad Rais[95], N'Goran G. Kouamé[96], Nieves García[97], Nono Legrand Gonwouo[98], Patricia A. Burrowes[99], Paul Y. Imbun[100], Philipp Wagner[101,102], Philippe J. R. Kok[103,104], Rafael L. Joglar[105,106], Renoir J. Auguste[107], Reuber Albuquerque Brandão[108], Roberto Ibáñez[109], Rudolf von May[110], S. Blair Hedges[111], S. D. Biju[112], S. R. Ganesh[113], Sally Wren[2,114], Sandeep Das[115,116], Sandra V. Flechas[117], Sara L. Ashpole[118,119], Silvia J. Robleto-Hernández[120], Simon P. Loader[38], Sixto J. Incháustegui[121], Sonali Garg[112,122], Somphouthone Phimmachak[123], Stephen J. Richards[124], Tahar Slimani[125], Tamara Osborne-Naikatini[126], Tatianne P. F. Abreu-Jardim[61], Thais H. Condez[127], Thiago R. De Carvalho[128], Timothy P. Cutajar[69,70], Todd W. Pierson[129], Truong Q. Nguyen[130], Uğur Kaya[44], Zhiyong Yuan[131], Barney Long[1], Penny Langhammer[1,132] & Simon N. Stuart[97,133,134]

[1]Re:wild, Austin, TX, USA. [2]IUCN SSC Amphibian Specialist Group, Toronto, Ontario, Canada. [3]Museu Paraense Emílio Goeldi, CZO/Herpetologia, Belém, Brazil. [4]Florida International University, Miami, FL, USA. [5]Centro de Ornitología y Biodiversidad (CORBIDI), Lima, Peru. [6]Laboratory of Animal Behaviour and Conservation, College of Life Sciences, Nanjing Forestry University, Nanjing, People's Republic of China. [7]Laboratory of Herpetology,

Museum Zoologicum Bogoriense, Research Center for Biosystematics and Evolution, National Research and Innovation Agency (BRIN), Cibinong, Indonesia. [8]Department of Zoology, Faculty of Science, Kasetsart University, Bangkok, Thailand. [9]Action Pour la Sauvegarde de l'Ecologie en Haïti (ACSEH), Les Cayes, Haiti. [10]Environmental Protection In the Caribbean (EPIC), Maho, Sint Maarten. [11]Museo de Vertebrados de la Universidad de Panamá, Ciudad de Panama, Panama. [12]Centro Oriental de Ecosistemas y Biodiversidad (BIOECO), Museo de Historia Natural "Tomás Romay", Santiago de Cuba, Cuba. [13]IUCN SSC Amphibian Specialist Group, Sri Lanka, Gampola, Sri Lanka. [14]Laboratoire Évolution & Diversité Biologique, UMR 5174, Université Toulouse III Paul Sabatier, Toulouse, France. [15]Russian State Agrarian University—MTAA, Moscow, Russia. [16]IUCN SSC Amphibian Specialist Group Bolivia, La Paz, Bolivia. [17]Animal Nutrition Unit, Department of Veterinary and Biosciences, Ghent University, Ghent, Belgium. [18]ASEAN Centre for Biodiversity, University of the Philippines Los Baños, Laguna, Philippines. [19]HerpWatch Pilipinas, Manila, Philippines. [20]Faculty of Education, University of the Ryukyus, Okinawa, Japan. [21]Graduate School of Engineering and Science, University of the Ryukyus, Okinawa, Japan. [22]SAVE THE FROGS!, Laguna Beach, CA, USA. [23]The University of Texas at Arlington, Arlington, TX, USA. [24]Smithsonian Conservation Biology Institute, Front Royal, VA, USA. [25]Center for Environmental Studies, Sanata Dharma University (CESSDU), Yogyakarta, Indonesia. [26]Pinelands Preservation Alliance, Southampton Township, NJ, USA. [27]Centro de Conservación de Anfibios, Amaru Bioparque, Cuenca, Ecuador. [28]Museo de Historia Natural, Escuela de Biologia, Universidad de San Carlos, Guatemala City, Guatemala. [29]FUNDAECO, Guatemala City, Guatemala. [30]Estación Biológica de Doñana (EBD-CSIC), Seville, Spain. [31]Department of Ecology and Environmental Sciences, Pondicherry University, Puducherry, India. [32]Universidade Federal de Mato Grosso, Cuiabá, Brazil. [33]Museo Nacional de Historia Natural, La Paz, Bolivia. [34]Sustainability Research Center & PhD Program in Conservation Medicine, Faculty of Life Sciences, Universidad Andres Bello, Santiago, Chile. [35]College of Science & Engineering, James Cook University, Townsville, Queensland, Australia. [36]David Attenborough Building, IUCN, Cambridge, UK. [37]Department of Life Sciences, University of the West Indies Mona, Kingston, Jamaica. [38]The Natural History Museum, London, UK. [39]Pacific Northwest Research Station, United States Department of Agriculture, Forest Service, Corvallis, OR, USA. [40]Universidad San Francisco de Quito USFQ, Colegio de Ciencias Biológicas y Ambientales, Instituto de Biodiversidad Tropical IBIOTROP, Quito, Ecuador. [41]Instituto Nacional de Biodiversidad INABIO, Quito, Ecuador. [42]Instituto de Biociências, Universidade Federal de Mato Grosso do Sul, Campo Grande, Brazil. [43]The New Guinea Binatang Research Center, Madang, Papua New Guinea. [44]Department of Zoology, Faculty of Science, Ege University, İzmir, Turkey. [45]Institute for Tropical Biology and Conservation, Universiti Malaysia Sabah, Kota Kinabalu, Malaysia. [46]Lee Kong Chian Natural History Museum, National University of Singapore, Singapore, Singapore. [47]Escuela de Biología, Universidad de Costa Rica, San José, Costa Rica. [48]CIBET (Museo de Zoología), Universidad de Costa Rica, San José, Costa Rica. [49]Chengdu Institute of Biology, Chinese Academy of Sciences, Chengdu, People's Republic of China. [50]Instituto de Investigación Biológica del Paraguay, Asunción, Paraguay. [51]Fundación Naturaleza El Salvador, San Salvador, El Salvador. [52]Museo Regionale di Scienze Naturali, Torino, Italy. [53]Zoologische Staatssammlung München (ZSM-SNSB), Munich, Germany. [54]Panthera, Tegucigalpa, Honduras. [55]Department of Ecology and Evolutionary Biology, University of Michigan, Ann Arbor, MI, USA. [56]Instituto de Biologia, Universidad Nacional Autónoma de México, Mexico City, Mexico. [57]Instituto de Ciencias Naturales, Universidad Nacional de Colombia, Bogotá D.C., Colombia. [58]WCS-Colombia, Cali, Colombia. [59]Biogeography and Spatial Ecology Research Group, Life Sciences Faculty, Universidad Regional Amazónica IKIAM, Tena, Ecuador. [60]Herpetology Division, Instituto Nacional de Biodiversidad, Quito, Ecuador. [61]Instituto Boitatá de Etnobiologia e Conservação da Fauna, Goiânia, Brazil. [62]Institute of Biodiversity and Environmental Conservation, Universiti Malaysia Sarawak, Kota Samarahan, Malaysia. [63]Departamento de Ciências Biológicas, Universidade Estadual de Santa Cruz, Ilhéus, Brazil. [64]Departamento de Ecología y Territorio, Facultad de Estudios Ambientales y Rurales, Pontificia Universidad Javeriana, Bogotá, Colombia. [65]Department of Biology and Ecology, Faculty of Sciences and Mathematics, University of Niš, Niš, Serbia. [66]Kadoorie Farm and Botanic Garden, Hong Kong SAR, People's Republic of China. [67]University of New England, Armidale, New South Wales, Australia. [68]Bhutan Ecological Society, Thimphu, Bhutan. [69]Australian Museum Research Institute, Australian Museum, Sydney, New South Wales, Australia. [70]Centre for Ecosystem Science, School of Biological, Earth and Environmental Sciences (BEES), University of New South Wales, Sydney, New South Wales, Australia. [71]Centre for Invasion Biology, Department of Botany & Zoology, Stellenbosch University, Stellenbosch, South Africa. [72]Centre for Invasion Biology, Institute of Biodiversity, School of Ecology and Environmental Science, Yunnan University, Kunming, People's Republic of China. [73]Laboratory for the Conservation of Endangered Species, CSIR-Centre for Cellular and Molecular Biology, Hyderabad, India. [74]Srishti Manipal Institute of Art, Design and Technology, Manipal Academy of Higher Education, Manipal, India. [75]Biolinx Environmental Research, Victoria, British Columbia, Canada. [76]Royal British Columbia Museum, Victoria, British Columbia, Canada. [77]Flint, TX, USA. [78]Laboratorio de Herpetología, Facultad de Ciencias, Universidad Nacional Autónoma de México, Mexico City, Mexico. [79]Laboratório de História Natural de Anfíbios Brasileiros (LaHNAB), Universidade Estadual de Campinas (Unicamp), São Paulo, Brazil. [80]Museo Nacional de Historia Natural de Cuba, La Habana, Cuba. [81]Department of Zoology, Jahangirnagar University, Dhaka, Bangladesh. [82]Key Laboratory in Forest Ecology and Conservation, College of Forestry, Guangxi University, Nanning, People's Republic of China. [83]Museo Nacional de Historia Natural "Jorge A. Ibarra", Ciudad de Guatemala, Guatemala. [84]Instituto de Biologia, Campus Universitário de Ondina, Universidade Federal da Bahia, Salvador, Brazil. [85]David, Panama. [86]Instituto de Ecorregiones Andinas (INECOA, UNJu—Conicet), San Salvador de Jujuy, Argentina. [87]Instituto Venezolano de Investigaciones Científicas (IVIC), Miranda, Venezuela. [88]Fundación para el Desarrollo de las Ciencias Físicas, Matemáticas y Naturales (FUDECI), Caracas, Venezuela. [89]Unidad de Investigación, Instituto Nacional de Biodiversidad (INABIO), Quito, Ecuador. [90]Natural History Museum of Denmark, University of Copenhagen, Copenhagen, Denmark. [91]Museum für Naturkunde—Leibniz Institute for Evolution and Biodiversity Science, Berlin, Germany. [92]Kyoto University, Yoshida Nihonmatsu, Kyoto, Japan. [93]Faculty of Forestry & Environment, IPB University, Bogor, Indonesia. [94]Aaranyak, Guwahati, India. [95]Herpetology Lab, Department of Zoology, Wildlife and Fisheries, Pir Mehr Ali Shah Arid Agriculture University Rawalpindi, Rawalpindi, Pakistan. [96]Laboratoire de Biodiversité et Ecologie Tropicale, UFR Environnement, Université Jean Lorougnon Guédé, Daloa, Côte d'Ivoire. [97]IUCN Species Survival Commission, Gland, Switzerland. [98]Laboratory of Zoology, Faculty of Science, University of Yaoundé I, Yaoundé, Cameroon. [99]Department of Biology, University of Puerto Rico, San Juan, Puerto Rico. [100]Zoology Unit, Research and Education Section, Sabah Parks, Kota Kinabalu, Malaysia. [101]Allwetterzoo, Münster, Germany. [102]Center for Biodiversity and Ecosystem, Villanova University, Villanova, PA, USA. [103]Department of Ecology and Vertebrate Zoology, Faculty of Biology and Environmental Protection, University of Łódź, Łódź, Poland. [104]Department of Life Sciences, The Natural History Museum, London, UK. [105]Rio Piedras Campus, University of Puerto Rico, San Juan, Puerto Rico. [106]Proyecto Coqui, San Juan, Puerto Rico. [107]Department of Life Sciences, The University of the West Indies, St Augustine, Trinidad and Tobago. [108]Laboratório de Fauna e Unidades de Conservação, Universidade de Brasília, Brasília-DF, Brazil. [109]Smithsonian Tropical Research Institute, Panama, República de Panamá. [110]California State University Channel Islands, Camarillo, CA, USA. [111]Center for Biodiversity, Temple University, Philadelphia, PA, USA. [112]Systematics Lab, Department of Environmental Studies, University of Delhi, Delhi, India. [113]Chennai Snake Park, Chennai, India. [114]Department of Zoology, University of Otago, Dunedin, New Zealand. [115]Centre for Research in Emerging Tropical Diseases, Department of Zoology, University of Calicut, Kerala, India. [116]EDGE of Existence programme, Conservation and Policy, Zoological Society of London, London, UK. [117]Bichos.team, Bogotá, Colombia. [118]Environmental Studies, St Lawrence University, Canton, NY, USA. [119]Prescott, Ontario, Canada. [120]IUCN SSC Amphibian Specialist Group Nicaragua, Managua, Nicaragua. [121]Grupo Jaragua, Santo Domingo, Dominican Republic. [122]Department of Organismic and Evolutionary Biology and Museum of Comparative Zoology, Harvard University, Cambridge, MA, USA. [123]Department of Biology, Faculty of Natural Sciences, National University of Laos, Vientiane, Laos. [124]Herpetology Department, South Australian Museum, Adelaide, South Australia, Australia. [125]Faculty of Sciences Sremlalia, Cadi Ayyad University, Marrakech, Morocco. [126]School of Agriculture, Geography, Environment, Ocean and Natural Sciences, The University of the South Pacific, Suva, Fiji. [127]Department of Earth Sciences, Carleton University, Ottawa, Ontario, Canada. [128]Universidade Federal de Minas Gerais, Belo Horizonte, Brazil. [129]Department of Ecology, Evolution and Organismal Biology, Kennesaw State University, Kennesaw, GA, USA. [130]Institute of Ecology and Biological Resources, Vietnam Academy of Science and Technology, Ha Noi, Viet Nam. [131]School of Life Sciences, Southwest University, Chongqing, People's Republic of China. [132]Arizona State University, Tempe, AZ, USA. [133]A Rocha International, London, UK. [134]Synchronicity Earth, London, UK. ✉e-mail: jluedtke@rewild.org

## Methods

### Data compilation

The Amphibian Red List Authority (ARLA) of the IUCN SSC Amphibian Specialist Group (ASG) coordinated the GAA2 according to the ASG's groupings of countries into regional working groups (Supplementary Table 2). Only a subset of the ASG regions was actively updating assessments at any one time.

Each regional assessment process addressed the endemic and non-endemic species in four stages: (1) pre-assessment; (2) expert consultation; (3) assessment finalizing and consistency checks; and (4) review. After the four stages were completed for all regions, the ARLA team retrospectively assigned a Red List category to all species for the years 1980 and 2004 (see the 'Backcasting Red List categories' section).

**Pre-assessment.** The GAA2 comprises reassessments of the 5,743 GAA1 species and the majority of species described and assessed for the first time between the two GAA projects (2004–2011). The GAA2 also contains an additional 2,286 newly described species assessed for the first time.

Regional species lists were compiled, incorporating taxonomic changes and new species descriptions collated by Amphibian Species of the World[40]. Literature reviews were conducted and any new published information was incorporated into draft assessments. In the case of reassessments, the newly available data were added to that of the previous assessment.

A particular challenge to this project is the dynamic state of amphibian taxonomy. By 2022, 191 of the GAA1 species had been synonymized, 24 were no longer considered valid species, three were considered hybrids and therefore ineligible for reassessment and four had been unintentionally assessed twice under different names.

**Expert consultation.** Over 1,000 subject-matter experts provided information to complete the required assessment fields (see the 'Extended acknowledgements' section in the Supplementary Information). A considerable amount of effort went into engaging with a diversity of experts across several axes (for example, gender, early versus late career researchers, geography, type of expertise) so as to reach the widest range of experts as possible and minimize reliance on any individual expert.

Future Global Amphibian Assessment initiatives would benefit from increasing the breadth of expertise engaged. Increased participation from conservation organizations and natural resource management or wildlife branches of governments should be targeted. Participants of both the first and second Global Amphibian Assessment were often members of academic institutions with expertise on herpetology, biogeography, taxonomy, and so on, as they were often the only scientists to have ever seen the species and visited known sites, and because they were typically experts in the species of the region or family of species being assessed. That said, participants without expertise in herpetology but with relevant expertise on regional threatening processes such as climate projections and wildlife trade, conservation planning, policy and implementation have the potential to improve the quality of the threat and conservation fields in the assessments.

Expert consultation of draft assessments was achieved through 31 in-person workshops, three remote workshops with over 180 online meetings, as well as phone and email correspondence (Supplementary Note 2). All workshops began with brief training in the IUCN categories and criteria, terms and definitions, and summary information from the *Guidelines for Using the IUCN Red List Categories and Criteria*[20] (IUCN Red List Guidelines). The online IUCN Red List Assessor Training Course[41] was made available ahead of workshops as an optional form of preparation, along with the *IUCN Red List Categories and Criteria*[42].

The expert consultation process was led by IUCN Red List trained facilitators and followed the IUCN Rules of Procedure[43]: (1) expert validation of the data in the assessments drafted during the pre-assessment stage. (i) In the early years of the GAA2 initiative, draft assessments were sent to experts for comment ahead of the data validation workshops. However, providing comments and data ahead of workshops quickly became infeasible due to the sheer number of species to be assessed. Thus, the preferred approach was for all data (both previous and new data) to be presented in sequential order to experts during workshops. (2) Contribution of missing data and/or revision of data with suitable justification. (i) In cases in which expert knowledge and/or unpublished data updated the information in the draft assessments, these were discussed and added during the workshop. (ii) Where possible, data quality was recorded using standardized data qualifiers (for example, observed, estimated, inferred, suspected) depending on the nature of evidence. Where no direct observational data were available, data fields (for example, population size and severity of threats) were derived through expert estimation or inference, according to 'Chapter 3: Data Quality' of the IUCN Red List Guidelines. Contributing experts were given an opportunity to comment or to revise any initial estimates, once they had a chance to discuss differences and to see the opinions of others. (3) Group discussion and application of the IUCN Red List Categories and Criteria to the data. (i) Uncertainty in the data and differences in risk tolerance between contributing experts were documented as a range of values in accordance with section 3.2.5 of the IUCN Red List Guidelines. When this resulted in a range of possible Red List categories being met (for example, Endangered–Critically Endangered), the range of categories was captured in the assessment rationale and a single category was chosen with clear justification for the decision, including whether an evidentiary or precautionary attitude was adopted. In cases in which the uncertainty was deemed to be too great, the category of Data Deficient was applied in compliance with section 10.3 of the IUCN Red List Guidelines. (ii) Of note are the differences in contribution between the workshop participants and workshop facilitators. The former brought expertise on the species and data relevant to the assessment, whereas the latter were experts in the IUCN categories and criteria. Thus, assessments were the product of both types of contributions.

We acknowledge that more formal elicitation methods, such as structured expert elicitation, can identify and reduce potential sources of bias and error among experts when contributing data and making judgements. This structured process could prove to be valuable for future IUCN Red List assessment processes, particularly for high-profile or contentious taxa, although it may be impractical for less-contentious taxa due to the amount of time required[44].

**Assessment finalizing.** The supporting data and Red List categories were finalized by an ARLA team member who also performed checks to ensure that the IUCN categories and criteria were applied in a consistent manner to the species within a particular region, but also between ASG regions. An example of an inconsistent result is when different Red List categories were determined for two or more species with very similar data. Consistency was also sought for species with similar traits or co-occurring species. If inconsistency was detected, assessments were revisited with data contributors to reconcile any discrepancies.

**Review.** An independent reviewer ensured biological accuracy and correct and consistent application of the Red List criteria. This process involved 15 independent reviewers between 2012 and 2022 (see the 'Extended acknowledgements' section in the Supplementary Information). The IUCN Red List Unit also reviewed assessments for appropriate application of the criteria.

### Data collected

Species assessments are required to meet the minimum documentation standards of the IUCN Red List as outlined in the Supporting Information Guidelines[45]. The supporting information includes information on

distribution, population, habitat preferences, ecology, use and trade, threats, conservation measures as well as the IUCN Red List category and criteria. Each assessment also includes a bibliography and the names of people involved in the process. This section describes the supporting data collected for each species.

**Systematics.** Higher taxonomy and scientific name, taxonomic authority, major synonyms, common names and taxonomic notes (if pertinent) were collected.

Occasionally, data from experts support an alternative taxonomic arrangement from that of the Amphibian Species of the World[40], which was accepted only in well-justified circumstances. Departures from Amphibian Species of the World are documented in the 'Taxonomic Notes' field of an assessment.

**Summary information.** Narrative texts about geographical range, population, habitat and ecology (including breeding and non-breeding habitats, as well as breeding strategy), threats, and conservation and research measures are required.

**Breeding strategy.** The breeding strategy of each amphibian was recorded in the IUCN Species Information Service on the basis of whether they (1) lay eggs; (2) give birth to live young; (3) exhibit parthenogenesis; (4) have a free-living larval stage; and/or (5) require water for breeding. When appropriate, the breeding strategy of a species was inferred from one or more congeners. Species were categorized as either larval developers, direct-developers, live-birth or unknown for the purpose of this study, as follows: larval developers (5,320 species): species coded as laying eggs and having a free-living larval stage. Direct developers (2,452 species): species coded as laying eggs but do not have a free-living larval stage. Live birth (61 species): species coded as giving birth to live young (viviparity) regardless of whether they have a free-living larval stage. Unknown (178 species): species coded as unknown for one or more questions, which prevented their breeding strategy from being categorized.

**Distribution map.** A map representing the currently known distribution of each species was generated according to the IUCN Mapping Standards[46]. The limits of a species' distribution were mapped using known occurrences of the taxon, and knowledge of habitat preferences, elevation limits and so on. Standard data attributes on presence, origin and seasonality were recorded for each range polygon. There are 53 species in the GAA2 without distribution maps as the taxon is known only from one or more specimens with no or extremely uncertain locality information.

**Additional distribution data.** Occurrences in biogeographic realms[47], biodiversity hotspots[48], countries and states or provinces (where required) were coded.

**Classification schemes.** To allow for comparative analyses and to ensure uniformity across species, a series of classification schemes[49] was used for habitats, threats, conservation actions, research needed, and use and trade.

**Red List category and criteria.** The IUCN Red List criteria were applied to the supporting data and the appropriate Red List category was determined, supported by a rationale[42]. A statement of the reason(s) for change in category from the previous assessment was documented for reassessed species. The date of assessment and the names of the facilitators, compilers and contributors were recorded.

**Backcasting Red List categories.** Only genuine changes in Red List category should be considered when comparing extinction risk in amphibians over time. A genuine change is either a real improvement or deterioration in the status of a species, driven by changes in the threat(s). For example, the protection of a species' habitat that halted the primary threat of deforestation could result in a genuine status improvement. On the other hand, a genuine status deterioration could be due to population declines associated with the introduction of a disease, the start of human activities causing ongoing habitat loss and degradation or the projected effects of climate change.

The majority of category changes from GAA1 to GAA2 were for non-genuine reasons. Generally, these were the result of the new information, such as distributional changes or clarity on threatening processes. For example, if a species was previously considered to be a narrow range endemic but was subsequently found to be much more widespread, the resulting change to a lower extinction risk category would be considered to be non-genuine. Other non-genuine reasons for category changes included changes in the application of the criteria or incorrect data used in the previous assessment(s).

A previous study[7] relied on the knowledge available at that time to backcast their 2004 assessments to 1980. This year corresponded approximately to the timeframe of severe population declines, as they were understood at the time. The GAA1 backcasted dataset provides a historical perspective taken into consideration in the GAA2 backcasting.

In early 2022, the ARLA team backcasted the GAA2 categories to 1980 and 2004 according to a method outlined previously[5]. This method uses the information in the Red List assessments in combination with additional knowledge on threatening processes, habitat decline trends and conservation actions (and in some cases further expert consultation) to determine whether a genuine change in a species' Red List category is likely to have occurred between 1980–2004 and 2004–2022. In the absence of notable evidence suggesting a genuine change, the GAA2 Red List category was assumed to be the same for previous time periods. Data Deficient species were automatically backcasted as data deficient in 1980 and 2004. Supplementary Table 3a,b contains the list of species that have deteriorated in status along with their backcasted categories, and Supplementary Table 4a,b contains the list of species that have improved in status.

**Primary drivers.** During the backcasting process, for species considered to have undergone a genuine category change since 1980, the relative importance of documented threats for each species was estimated. The most notable perceived threat was assigned as the 'primary driver' and selected from the following list: agriculture, mining/energy production, infrastructure development, human disturbance, timber and plant harvesting, anthropogenic fire, water management, native species, introduced species, pollution, geological events, disease, overexploitation, climate change effects and undetermined.

Species that deteriorated in status were assigned the primary driver that contributed to the category change. For species that improved in status, the primary driver that was previously causing the deterioration but has since been mitigated were assigned. Improvements that were the result of conservation action were documented through an additional data field (Supplementary Tables 3a,b and 4a,b).

## Data limitations

**Regional variation.** IUCN Red List assessments are considered to be out of date 10 years after the date of assessment. Thus, all species included in the GAA2 have been assessed within the past ten years and are considered current. However, for regions that were assessed earlier in the GAA2, the data are comparatively less current than for the regions completed during the latter stages of the project.

For example, towards the end of the GAA2, the severity, scope and timing of the effects of climate change were at the forefront of discussions but were not as well addressed for earlier regions. Thus, the species- and habitat-specific effects of climate change are probably underestimated for regions that were assessed earlier in the GAA2.

Data scarcity was a common issue for regions with few herpetologists and for species occurring in areas that are difficult to access. As such, assessments in data-poor regions, such as Melanesia and sub-Saharan Africa, generally contain substantially less detail compared with data-rich regions such as North America, Australia and Europe, where species are often relatively well studied. This is also true for population data, where there has been little (if any) population monitoring, and threat-determining processes with scarce published literature on climate change, rates of habitat loss or exploitation.

The rate of new species descriptions also varies regionally, with the amphibian fauna in many parts of the world still very poorly known. Thus, the currently known amphibian richness and diversity is substantially underestimated in those places.

**Not evaluated species.** The GAA2 aimed to assess the extinction risk of all taxonomically valid amphibian species. However, as the annual rate of new species descriptions remains high, inevitably some newly described species are not included in the GAA2. After a region had been completed during the GAA2, all subsequent new species descriptions for that region were reserved for the GAA3. On occasion, a few species were assessed after the Red List update for a region was no longer active—typically when a species was known to be facing serious threats or there were taxonomic implications for regions that were actively being updated. As of December 2022, the number of new species waiting to be assessed in the GAA3 was approaching 400 and is steadily increasing as new species descriptions are published weekly.

**Data Deficient species.** In the GAA2, 909 species were categorized as data deficient owing to insufficient data. At a minimum, Data Deficient species are expected to be threatened at a similar proportion as the global average of threatened species (40.7%). Owing to these data gaps, we expect the number of genuine changes to also be underestimated. This may be the case for Data Deficient species that have not been surveyed for decades and for which there is no information to confirm whether population declines have taken place.

### Analytical methods
**Percentage of threatened species.** Species in the Critically Endangered (CR), Endangered (EN) and Vulnerable (VU) categories are referred to as threatened species.

When determining the percentage of threatened species in this study, a best estimate was calculated excluding the number of Data Deficient (DD) and Extinct (EX) species from the total. However, Extinct in the Wild (EW) species were included because there remains the possibility that they can be reintroduced to the wild. To capture the uncertainty within this estimate, a lower estimate was calculated by assuming that all Data Deficient species are not threatened, and an upper estimate is calculated by assuming that all Data Deficient species are threatened:

$$\text{Lower estimate} = (EW + CR + EN + VU)/(\text{total species} - EX)$$

$$\text{Best estimate} = (EW + CR + EN + VU)/(\text{total species} - EX - DD)$$

$$\text{Upper estimate} = (EW + CR + EN + VU + DD)/(\text{total species} - EX)$$

For further details and discussion of these methods, see the IUCN Red List Resources Summary Statistics documentation[50].

**Threats to threatened species.** The GAA2 coded threats affecting amphibians using the threat-classification scheme (see the 'Classification schemes' section). When relevant, more than one threat was coded per species. The timing of the threat (past, ongoing, future), and the resulting stresses to the species, were also indicated.

In Fig. 2, the hierarchy within the threat-classification scheme was used to group similar threats and allow for comparison, although some, such as *B. dendrobatidis*, were separated to highlight their significance. Only ongoing and future major threats to threatened species are included. To highlight the emerging nature of *B. dendrobatidis*, *B. salamandrivorans* and climate change effects, the number of threatened species for which these factors are only a future threat are indicated by hatching on the bars.

Threat groupings were as follows:
- Agriculture: all codes under 2 Agriculture & aquaculture.
- Timber and plant harvesting: all codes under 5.2 Gathering terrestrial plants and 5.3 Logging and wood harvesting.
- Infrastructure development: all codes under 1 Residential & commercial development and 4 Transportation & service corridors.
- Pollution: all codes under 9 Pollution.
- Mining/energy production: all codes under 3 Energy production & mining.
- Water management: all codes under 7.2 Dams & water management.
- Human disturbance: all codes under 6 Human intrusions & disturbance.
- Geological events: all codes under 10 Geological events.
- Over-exploitation: all codes under 5.1 Hunting & collecting terrestrial animals and 5.4 Fishing & harvesting aquatic resources.
- Climate change: all codes under 11 Climate change & severe weather.
- Fire: all codes under 7.1 Fire & fire suppression.
- *B. dendrobatidis*: under the codes 8.1.2 Invasive non-native/alien species/diseases—named species and 8.4.2 Problematic species/diseases of unknown origin—named species, the name of invasive/problematic species must be recorded. Only records for which *B. dendrobatidis* was listed were included.
- *B. salamandrivorans*: under the codes 8.1.2 Invasive non-native/alien species/diseases—named species and 8.4.2 Problematic species/diseases of unknown origin—named species, the name of invasive/problematic species must be recorded. Only records for which *B. salamandrivorans* was listed were included.
- Invasive species: all codes under 8.1 Invasive non-native/alien species/diseases, 8.3 Introduced genetic material, 8.4 Problematic species/diseases of unknown origin, 8.5 Viral/prion-induced diseases and 8.6 Diseases of unknown cause, except when the invasive/problematic species is identified as *B. dendrobatidis* or *B. salamandrivorans*.
- Native species: all codes under 8.2 Problematic native species/diseases.

**RLI.** Determining trends in the extinction risk of amphibians requires that only genuine changes in the Red List category between assessments be included in the RLI. Thus, the backcasted 1980 and 2004 categories assigned in the GAA2 (Extended Data Table 1; see the 'Backcasting red list categories' section) are used to calculate the RLI for amphibians.

The RLI is calculated according to the methods outlined previously[5] and detailed online[51]. The value of the RLI at each datapoint is an indication of the average extinction risk of all species at that point in time and can range from 0 (all species are Extinct) to 1 (all species are Least Concern). The gradient (slope) of the line is a measure of the rate of change in Red List categories. Thus, a steep negative gradient would indicate that a considerable proportion of species moved from a less threatened to a more threatened Red List category. By contrast, a positive gradient is indicative of an overall improvement.

Note that CR(PE) and EX species are weighted the same when calculating the RLI. Thus, a change in category from CR(PE) to EX from one time period to the next is not considered to be a deterioration in status; however, a change from CR to CR(PE) is treated as such. Data Deficient species are not included in the RLI as their extinction risk is still unknown.

The RLIs for other comprehensively assessed taxonomic groups are included in Fig. 2a to allow for a direct comparison with amphibians. The relatively small number of amphibians (264) occurring across more

than one biogeographical realm were included in the disaggregated RLI calculations of each realm of occurrence (Fig. 3b). This is considered to be the best approach for representing the overall extinction risk of a given realm.

The decline in the amphibian RLI could initially be interpreted as minimal. However, to put this trend into perspective, 482 amphibians moved into a higher extinction risk category between 2004 and 2022 and 306 between 1980 and 2004 (Extended Data Table 2).

**Grouping of primary drivers.** For species that changed categories between assessment periods, a primary driver responsible for the change was allocated (see the 'Primary drivers' section; Supplementary Table 3a,b). Many of these primary drivers cause habitat loss and degradation. For the purpose of this study, the drivers were further grouped as follows:

- Habitat loss/degradation: agriculture, mining/energy production, infrastructure development, human disturbance, timber and plant harvesting, anthropogenic fire, water management, native species, pollution, geological events.
- Disease: chytridiomycosis only.
- Over-exploitation: over-exploitation only.
- Climate change effects: climate change effects only.
- Undetermined: includes a small number of species for which there is insufficient information regarding what is/are the driver(s) of the change in category.
- Numerous: includes a small number of species (5) that have more than one driver that are considered to be contributing equally to the change in category.

Invasive species are documented as a threat to 415 threatened species (Fig. 2). However, except for the species that are probably affected by the amphibian chytrid fungus, *B. dendrobatidis*, no amphibians in this study experienced a deterioration in status due to invasive non-native species. A small number of category changes were driven by the threats native species, geological events and anthropogenic fire, which cause habitat degradation and were therefore grouped under habitat loss/degradation.

Over-exploitation was the primary driver for 31 status deteriorations during 1980–2004 compared with only 4 during 2004–2022 (Extended Data Table 2). Deteriorations in status due to over-exploitation remain concentrated in Indomalaya (Extended Data Table 3), particularly in eastern and southeastern Asia (Fig. 4). However, population declines due to over-exploitation are typically based on expert opinion because very little data exist on utilization rates of amphibians. As a result, it was often difficult to accurately determine when and to what degree a species deteriorated in status.

## Reporting summary

Further information on research design is available in the Nature Portfolio Reporting Summary linked to this article.

## Data availability

The spatial and raw tabular data analysed in this study are available online (https://www.iucnredlist.org/resources/data-repository). The GAA2 IUCN Red List assessments, including range maps, for all 8,011 species will be available for download on The IUCN Red List of Threatened Species website (https://iucnredlist.org) after its December 2023 update (version 2023–2). In rare cases, a species may be threatened because of over-collection and sensitive distribution information is not publicly available. Source data are provided with this paper.

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

**Acknowledgements** We thank each of the contributors named in the Supplementary Information and D. Church, W. Sechrest, P. Ghosh, J. P. Rodriguez, K. Mileham and R. Akcakaya, without whom the GAA2 would have not been possible. The views expressed here do not necessarily reflect those of IUCN's staff, members or commissions. The designation of geographical entities in this paper, and the presentation of the material, do not imply the expression of any opinion whatsoever on the part of IUCN concerning the legal status of any country, territory or area, or of its authorities, or concerning the delimitation of its frontiers or boundaries. The majority of funds were provided by Re:wild, Synchronicity Earth, Kering and the Environment Agency—Abu Dhabi Framework Grant to the IUCN Species Survival Commission (S.N.S., J.A.L., P.L. and B.L.). Grants for specific components of the GAA2 are listed as follows. Alashan SEE Foundation grant The IUCN Red List Workshop for Chinese Amphibian Species (to J.A.L. and L.H.). Detroit Zoological Society grant Catalyzing Amphibian Conservation in Honduras (to J.A.L., K.N., L.H. and P.L.). Detroit Zoological Society grant Support to the IUCN SSC Amphibian Red List Authority for the Updates of Madagascar and the Tropical Americas (to J.A.L. and A. Angulo). Dilmah Conservation grant Updating the IUCN Red List Assessments for the Amphibians of Sri Lanka (to J.A.L., K.N., L.H. and P.L.). Honolulu Zoo grant Support to Amphibian Conservation in Southeast Asia (to J.A.L. and A. Angulo). IUCN Red List Committee allocation of funds raised in 2015, 2016, 2017, 2018, 2019 and 2020 through the Integrated Biodiversity Assessment Tool (IBAT) to the IUCN SSC Amphibian Specialist Group Red List Authority (to J.A.L., S.N.S., P.L., B.L., K.N. and L.H.). Kering grant Critical Biodiversity Information for Decision-Making (to J.A.L., P.L., K.N. and L.H.). Museo delle Scienze di Trento to the IUCN SSC Amphibian Specialist Group Red List Authority (to J.A.L. and A. Angulo). NatureServe grant Andean Species-level Indicators (to J.A.L., K.N. and L.H.). Rainforest Trust grant Identifying Priority Sites for the Most Threatened Amphibian Species (to J.A.L., K.N., L.H. and S.N.S.). Yayasan Belantara grant support for the Indonesian Amphibian Red List Assessment Workshop (to M.D.K. and A.H.). In addition to the many institutions that hosted workshops and provided the time of their staff to participate in the GAA2, we recognize the contributions of the following organizations: the American Museum of Natural History's Amphibian Species of the World; Re:wild; the IUCN Biodiversity Assessment Unit; the Amphibian Survival Alliance; the IUCN SSC Chair's Office; the Red List Technical Working Group; Amphibian Ark; NatureServe; Synchronicity Earth; and iNaturalist. Serving as the gatekeeper to the Red List, the members of the IUCN Red List Unit provided technical training and support, maintenance of the online Species Information Service database, assessment checks and publication of assessments on the Red List website. We recognize in particular the contributions of A. Joolia, A. M. Richardt, C. Pollock, C. Hilton-Taylor, J. Scott and J. Window. The GAA2 was an extensive collaboration of over 1,000 individuals. We acknowledge their contributions and have listed their contributions individually in the 'Extended acknowledgements' section of the Supplementary Information.

**Author contributions** Conceptualization: J.A.L., J.C., K.N., L.H., B.L., P.L. and S.N.S. Methodology: J.A.L., J.C., K.N., L.H., C.H.-T., B.L., P.L. and S.N.S. Validation: J.A.L., J.C., K.N., L.H., C.H.-T., B.L., P.L. and S.N.S. Formal analysis: J.C., K.N. and C.H.-T. Investigation: J.A.L., J.C., K.N., L.H., A.A.K., A. Angulo, A. Aowphol, A.B., A.C., A.C.D., A.F., A.F.G., A.H., A.J., A.M.S., A.O.M., A.d.S., A.S.-B., A.T., B.G., B.S., B.T., C.A., C.C.M.R., C.F.C.F., C.H.-T., C.J.H., C.R.V.A., C. Señaris, C.S.R., C. Strüssmann, D.F.C.-H., D.H.O., D.J.G., D.J.S., D.L.W., E.N., E.N.-M., E.S.H.Q., F.A., F. Bolaños, F. Brusquetti, F.E.C., F.G., F.K., F.S.Á., F.X., G.C., G.F.M.-R., G.G.-D., G.P.-O., H.M.O.-A., I.D., I.F.M., I.R.D., J.C.-I., J.-H.Y., J.J., J.J.L.R., J.M., J.N.U.-C., J.T.W., K.O., K.O.C., K.V., K.V.G., L.C.-M., L.C.W., L.F.T., L.M.D., M.A.P., M.D.K., M.E.A., M.F., M.F.A., M.F.N., M.H.Y.-M., M.L., M. Matsui, M. Meegaskumbura, M.M.H.K., M.-O.R., M.R., M.D.S., M.V., N.G., N.G.G.K., N.L.G., P.A.B., P.J.R.K., P.W., P.Y.I., R.A.B., R.I., R.J.A., R.L.J., R.v.M., S.B.H., S.D., S.D.B., S.G., S.J.I., S.J.R., S.J.R.-H., S.L.A., S.P., S.P.L., S.R.G., S.V.F., S.W., T.H.C., T.O.-N., T.P.C., T.P.F.A.-J., T.Q.N., T.R.D.C., T.S., T.W.P., U.K. and Z.Y. Data curation: JAL, JC, KN, LH, C.H.-T., EN, LCW, NG, T.P.F.A.-J. Writing—original draft: J.A.L., J.C., K.N., L.H., B.L., P.L. and S.N.S. Writing—review and editing: J.A.L., J.C., K.N., L.H., A.A.K., A. Angulo, A. Aowphol, A.B., A.C., A.C.D., A.F., A.F.G., A.H., A.J., A.M.S., A.O.M., A.d.S., A.S.-B., A.T., B.G., B.S., B.T., C.A., C.C.M.R., C.F.C.F., C.H.-T., C.J.H., C.R.V.A., C. Señaris, C.S.R., C. Strüssmann, D.F.C.-H., D.H.O., D.J.G., D.J.S., D.L.W., E.N., E.N.-M., E.S.H.Q., F.A., F. Bolaños, F. Brusquetti, F.E.C., F.G., F.K., F.S.Á., F.X., G.C., G.F.M.-R., G.G.-D., G.P.-O., H.M.O.-A., I.D., I.F.M., I.R.D., J.C.-I., J.-H.Y., J.J., J.J.L.R., J.M., J.N.U.-C., J.T.W., K.O., K.O.C., K.V., K.V.G., L.C.-M., L.C.W., L.F.T., L.M.D., M.A.P., M.D.K., M.E.A., M.F., M.F.A., M.F.N., M.H.Y.-M., M.L., M. Matsui, M. Meegaskumbura, M.M.H.K., M.-O.R., M.R., M.D.S., M.V., N.G., N.G.G.K., N.L.G., P.A.B., P.J.R.K., P.W., P.Y.I., R.A.B., R.I., R.J.A., R.L.J., R.v.M., S.B.H., S.D., S.D.B., S.G., S.J.I., S.J.R., S.J.R.-H., S.L.A., S.P., S.P.L., S.R.G., S.V.F., S.W., T.H.C., T.O.-N., T.P.C., T.P.F.A.-J., T.Q.N., T.R.D.C., T.S., T.W.P., U.K., Z.Y., B.L., P.L. and S.N.S. Visualization: J.C. and K.N. Supervision: J.A.L., J.C., C.H.-T., B.L., P.L. and

S.N.S. Project administration: J.A.L., J.C., K.N., L.H., A. Angulo, A.C., A.C.D., A.M.S., A.d.S., A.T., C.A., C.C.M.R., C.F.C.F., C.R.V.A., C. Señaris, D.F.C.-H., D.H.O., E.N.-M., F.A., F.E.C., F.G., F.X., G.C., G.P.-O., I.F.M., J.C.-I., J.J., J.J.L.R., J.M., J.N.U.-C., J.T.W., K.O., K.O.C., K.V., K.V.G., M.D.K., M.L., M. Matsui, M. Meegaskumbura, M.-O.R., M.D.S., M.V., N.L.G., R.I., S.B.H., S.J.I., S.J.R., S.P.L., S.W., T.P.C., T.W.P. and U.K. Funding acquisition: J.A.L., K.N., L.H., A.C.D., F.A., J.J.L.R., J.M., M.D.K., B.L., P.L. and S.N.S.

**Competing interests** The authors declare no competing interests.

**Additional information**
**Correspondence and requests for materials** should be addressed to Jennifer A. Luedtke.

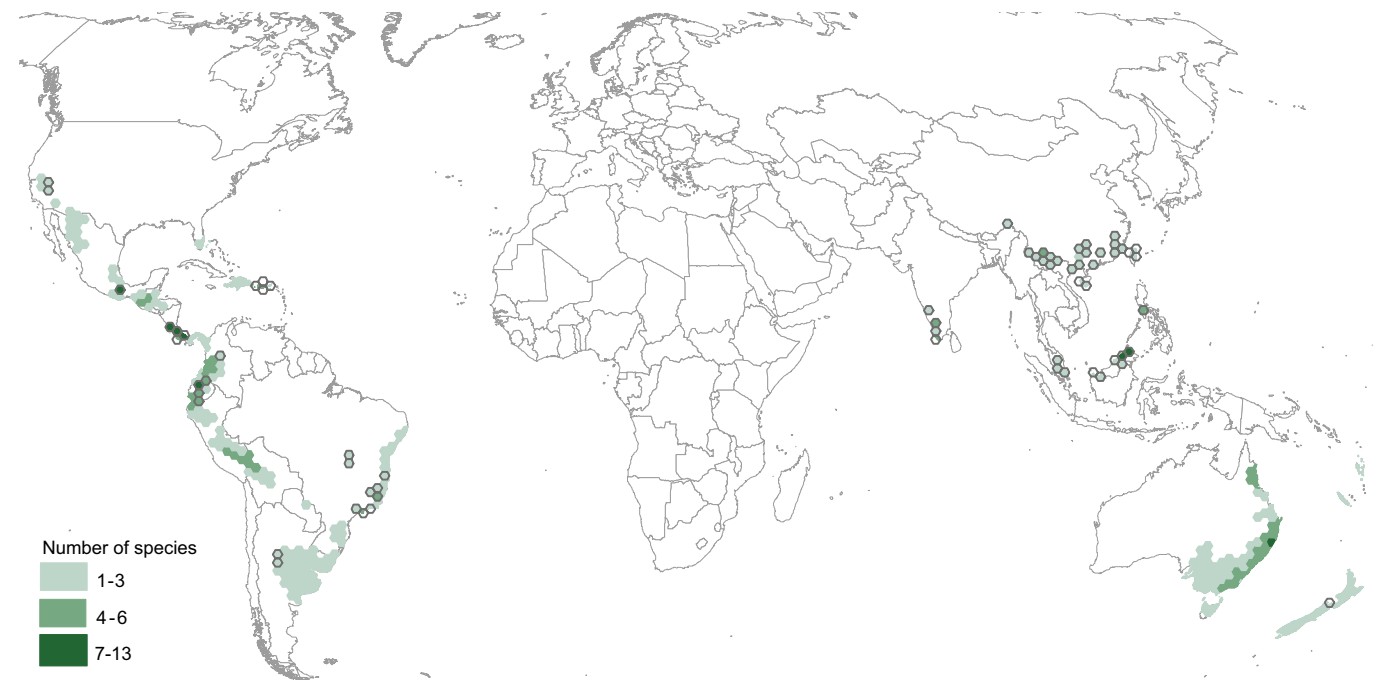

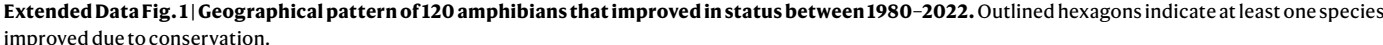

**Extended Data Fig. 1 | Geographical pattern of 120 amphibians that improved in status between 1980–2022.** Outlined hexagons indicate at least one species improved due to conservation.

**Extended Data Table 1 | Number of species in each Red List category for 1980, 2004, and 2022**

| Red List Category | 1980 | 2004 | 2022 |
|---|---|---|---|
| EX | 23 | 33 | 37 |
| EW | 0 | 1 | 2 |
| CR(PEW) | 0 | 1 | 1 |
| CR(PE) | 24 | 162 | 185 |
| CR | 564 | 604 | 612 |
| EN | 1,293 | 1,235 | 1,264 |
| VU | 800 | 786 | 811 |
| NT | 384 | 413 | 451 |
| LC | 4,014 | 3,868 | 3,739 |
| DD | 909 | 909 | 909 |
| **Percentage of Species Threatened** | | | |
| Lower estimate | 33.6 | 35.0 | 36.1 |
| Best estimate | 37.9 | 39.4 | 40.7 |
| Upper estimate | 44.9 | 46.3 | 47.5 |

The 1980 and 2004 categories were determined by applying the backcasting methods outlined in Butchart et al.[5]. The 2022 Red List categories are the results of the GAA2 study and the most recent assessment for each species. The Critically Endangered (CR) category has an additional option to tag a species as "Possibly Extinct (PE)" or "Possibly Extinct in the Wild (PEW)". The disaggregation of CR species has been provided in this table to emphasize the large number of amphibians that are categorised as CR(PE). Following the methods outlined in Section 4.1, the best, lower, and upper estimate of the percentage of threatened or extinct species is calculated for each point in time. There has been a steady increase in the percentage of threatened amphibians from 37.9% (1980) to 39.4% (2004) to 40.7% (2022). It should be noted that the two time periods (1980–2004 and 2004–2022) are not equal; the first one being 24 years and the second only 18. From 1980 to 2004, an additional 118 species were categorised as threatened. An additional 90 species are threatened as of 2022. From 1980 to 2004, the total number of species listed as VU and EN decreased, while the number listed as CR considerably increased from 588 to 766. In 1980, 24 species were considered CR(PE), but by 2004 the number of CR(PE) species rose to 162. The number of species declared EX also increased from 23 in 1980, to 33 in 2004. In contrast, from 2004 to 2022, the number of species in each of the threatened categories increased by a similar amount; the number of CR(PE) species increased by 23; and the number of EX species increased by four, which is substantially less than the previous time period, but still of significant conservation concern.

**Extended Data Table 2 | Species with status deteriorations in each time period (1980–2004 and 2004–2022)**

| Grouped driver | Primary driver | 1980–2004 | 2004–2022 |
|---|---|---|---|
| Disease | | **281** | **69** |
| Habitat loss/ degradation | | **156** | **112** |
| | Agriculture | 93 | 57 |
| | Mining/energy production | 22 | 27 |
| | Infrastructure development | 19 | 11 |
| | Timber and plant harvesting | 9 | 1 |
| | Water management | 8 | 4 |
| | Human disturbance | 2 | 2 |
| | Anthropogenic fire | 1 | 5 |
| | Pollution | 1 | 1 |
| | Geological events | 1 | 1 |
| | Native species | 0 | 3 |
| Over-exploitation | | **31** | **4** |
| Climate change effects | | **6** | **119** |
| Undetermined | | **6** | **1** |
| Numerous | | **2** | **1** |
| **Total** | | **482** | **306** |

Species are categorised by the primary driver of the status deterioration. Primary drivers are grouped in the first column and separated in the second.

**Extended Data Table 3 | Number of species with status deteriorations in each time period (1980–2004 and 2004–2022) disaggregated by the data groupings used to calculate the Red List Indices and primary drivers of status deteriorations**

| Data grouping | Time period | Primary driver | | | | |
|---|---|---|---|---|---|---|
| | | Disease | Habitat loss/ degradation | Over-exploitation | Climate change effects | Numerous |
| Afrotropics | 1980–2004 | 3 | 12 | 1 | 0 | 0 |
| | 2004–2022 | 11 | 3 | 0 | 4 | 0 |
| Australasia/ Oceania | 1980–2004 | 25 | 4 | 0 | 4 | 0 |
| | 2004–2022 | 2 | 8 | 0 | 11 | 0 |
| Indomalaya | 1980–2004 | 0 | 3 | 9 | 0 | 0 |
| | 2004–2022 | 0 | 5 | 1 | 0 | 1 |
| Nearctic | 1980–2004 | 4 | 10 | 0 | 1 | 0 |
| | 2004–2022 | 0 | 5 | 0 | 12 | 0 |
| Neotropics | 1980–2004 | 250 | 114 | 4 | 1 | 2 |
| | 2004–2022 | 45 | 77 | 0 | 91 | 0 |
| Palaearctic | 1980–2004 | 0 | 26 | 12 | 0 | 0 |
| | 2004–2022 | 11 | 16 | 2 | 1 | 0 |
| Larval developer | 1980–2004 | 203 | 107 | 31 | 1 | 2 |
| | 2004–2022 | 51 | 65 | 4 | 42 | 1 |
| Direct developer | 1980–2004 | 76 | 45 | 0 | 5 | 0 |
| | 2004–2022 | 15 | 47 | 0 | 73 | 0 |
| Live birth | 1980–2004 | 1 | 3 | 0 | 0 | 0 |
| | 2004–2022 | 3 | 0 | 0 | 2 | 0 |
| Anura | 1980–2004 | 272 | 139 | 23 | 5 | 2 |
| | 2004–2022 | 57 | 89 | 1 | 103 | 1 |
| Caudata | 1980–2004 | 9 | 15 | 8 | 1 | 0 |
| | 2004–2022 | 12 | 22 | 3 | 16 | 0 |
| Gymnophiona | 1980–2004 | 0 | 2 | 0 | 0 | 0 |
| | 2004–2022 | 0 | 1 | 0 | 0 | 0 |

In the Neotropics, disease stands out as by far the most common driver of status deteriorations between 1980–2004 (250 species), but this driver diminished between 2004–2022 (45 species). Climate change effects were only implicated for one species in the Neotropics between 1980–2004 but increased substantially to 91 species between 2004–2022. A similar trend is shown in the Nearctic and Australasia/Oceania. Interestingly, the Afrotropical region shows the reverse trend for disease, with the number of species deteriorating in status increasing from three in the first time period to 11 in the second, due to recent *Bd* outbreaks emerging in central and eastern Africa. In the Palaearctic, the increasing impact of disease is also noticeable, and can be attributed to the recent introduction of *Bsal* and the impact its predicted spread will have on many salamanders. For Anura, the impact of disease has greatly diminished with time, and climate change effects have more recently emerged as the most common primary driver, although habitat loss/degradation is still prominent. With the emergence of *Bsal*, disease has remained an overall concern for Caudata, although climate change effects are now also considered the most common primary driver. The trend of diminishing impacts due to disease in the first period, and the emergence of climate change effects in the second period seems to be similar for both larval and direct developers.

# Reporting Summary

## Statistics

For all statistical analyses, confirm that the following items are present in the figure legend, table legend, main text, or Methods section.

| n/a | Confirmed | |
|---|---|---|
| ☐ | ☒ | The exact sample size (*n*) for each experimental group/condition, given as a discrete number and unit of measurement |
| ☒ | ☐ | A statement on whether measurements were taken from distinct samples or whether the same sample was measured repeatedly |
| ☒ | ☐ | The statistical test(s) used AND whether they are one- or two-sided *Only common tests should be described solely by name; describe more complex techniques in the Methods section.* |
| ☒ | ☐ | A description of all covariates tested |
| ☒ | ☐ | A description of any assumptions or corrections, such as tests of normality and adjustment for multiple comparisons |
| ☐ | ☒ | A full description of the statistical parameters including central tendency (e.g. means) or other basic estimates (e.g. regression coefficient) AND variation (e.g. standard deviation) or associated estimates of uncertainty (e.g. confidence intervals) |
| ☒ | ☐ | For null hypothesis testing, the test statistic (e.g. *F*, *t*, *r*) with confidence intervals, effect sizes, degrees of freedom and *P* value noted *Give P values as exact values whenever suitable.* |
| ☒ | ☐ | For Bayesian analysis, information on the choice of priors and Markov chain Monte Carlo settings |
| ☒ | ☐ | For hierarchical and complex designs, identification of the appropriate level for tests and full reporting of outcomes |
| ☒ | ☐ | Estimates of effect sizes (e.g. Cohen's *d*, Pearson's *r*), indicating how they were calculated |

*Our web collection on statistics for biologists contains articles on many of the points above.*

## Software and code

Policy information about availability of computer code

| Data collection | No software was used. |
|---|---|
| Data analysis | No software was used. |

For manuscripts utilizing custom algorithms or software that are central to the research but not yet described in published literature, software must be made available to editors and reviewers. We strongly encourage code deposition in a community repository (e.g. GitHub). See the Nature Portfolio guidelines for submitting code & software for further information.

## Data

Policy information about availability of data

All manuscripts must include a data availability statement. This statement should provide the following information, where applicable:
- Accession codes, unique identifiers, or web links for publicly available datasets
- A description of any restrictions on data availability
- For clinical datasets or third party data, please ensure that the statement adheres to our policy

The spatial and raw tabular data analysed in this study are available at https://doi.org/10.5061/dryad.xgxd254n5.
The GAA2 IUCN Red List assessments, including range maps, for all 8,011 species will also be available for download on The IUCN Red List of Threatened SpeciesTM website (https://iucnredlist.org) following its September 2023 update (version 2023–1).

In rare cases, a species may be threatened because of over-collection and sensitive distribution information is not publicly available. Source data are provided with this paper.

# Research involving human participants, their data, or biological material

Policy information about studies with human participants or human data. See also policy information about sex, gender (identity/presentation), and sexual orientation and race, ethnicity and racism.

| | |
|---|---|
| Reporting on sex and gender | N/A |
| Reporting on race, ethnicity, or other socially relevant groupings | N/A |
| Population characteristics | N/A |
| Recruitment | N/A |
| Ethics oversight | N/A |

Note that full information on the approval of the study protocol must also be provided in the manuscript.

# Field-specific reporting

Please select the one below that is the best fit for your research. If you are not sure, read the appropriate sections before making your selection.

☐ Life sciences    ☐ Behavioural & social sciences    ☒ Ecological, evolutionary & environmental sciences

For a reference copy of the document with all sections, see nature.com/documents/nr-reporting-summary-flat.pdf

# Ecological, evolutionary & environmental sciences study design

All studies must disclose on these points even when the disclosure is negative.

| | |
|---|---|
| Study description | This study examines the 8,011 amphibian species with an extinction risk assessment for the IUCN Red List of Threatened Species. Trends in extinction risk are quantified for 1980, 2004, and 2022 with comparisons between species in the different biogeographic realms, taxonomic orders, and breeding strategies. Estimates of extinction risk using current data are made for the species that were not known to science in 1980 and 2004. A particular focus of the study is the drivers of genuine extinction risk changes as these reflect actual increases or decreases in threat levels, some due to targeted conservation actions. These results are relevant to global, national, and local conservation planning and prioritisation, the National Biodiversity Action Plans (NBSAPs) reported to the Convention on Biological Diversity (CBD) of the United Nations to track progress towards the Kunming-Montreal Global Biodiversiy Framework adopted by 190+ signatory countries at COP15 in Montreal, Canada in December 2022. |
| Research sample | The sample size of this study includes 8,011 amphibian species known to science, representing 92.9% of described amphibians on the 3 May 2023 resubmission date. |
| Sampling strategy | The entire sample was used. |
| Data collection | Raw data collection took place between 2012-2022 resulting in IUCN Red List categories and their accompanying information for each species. This information comprises one of the two datasets in this study. This process involved more than 1,000 subject-matter experts through the consultation process described in the Methods section of the manuscript. Backcasting of the categories took place in 2022, which comprises the second dataset analysed in this study. |
| Timing and spatial scale | Data collection took place between 2012-2022. The data cover the taxonomy and geographic range of the 8,011 amphibian species in this study, i.e. every continent except Antarctica. |
| Data exclusions | No data were excluded. |
| Reproducibility | The data made available in the manuscript, Supplementary Information, and the data repository linked above enable the reproduction of all analyses and results. |
| Randomization | No randomisation was necessary for the analyses inf this study. |
| Blinding | Blinding was not relevant to this study. |

Did the study involve field work?    ☐ Yes    ☒ No

nature portfolio | reporting summary

# Reporting for specific materials, systems and methods

We require information from authors about some types of materials, experimental systems and methods used in many studies. Here, indicate whether each material, system or method listed is relevant to your study. If you are not sure if a list item applies to your research, read the appropriate section before selecting a response.

## Materials & experimental systems

| n/a | Involved in the study |
|-----|----------------------|
| ☒ ☐ | Antibodies |
| ☒ ☐ | Eukaryotic cell lines |
| ☒ ☐ | Palaeontology and archaeology |
| ☒ ☐ | Animals and other organisms |
| ☒ ☐ | Clinical data |
| ☒ ☐ | Dual use research of concern |
| ☒ ☐ | Plants |

## Methods

| n/a | Involved in the study |
|-----|----------------------|
| ☒ ☐ | ChIP-seq |
| ☒ ☐ | Flow cytometry |
| ☒ ☐ | MRI-based neuroimaging |

