## [Peer Review File · Nature]

Manuscript Title: Ongoing declines for the world's amphibians in the face of emerging threats

Reviewer Comments & Author Rebuttals

Reviewer Reports on the Initial Version:

Referees' comments:

Referee #1 (Remarks to the Author):

This manuscript presents the results of a very considerable body of work, presenting data and analyses of international significance for amphibian ecology and the mitigation of biodiversity loss. The analysis rests on the data and judgements of a large number of professionals and the well-established IUCN Red List protocols and it has used them in a robust and conventional way.

However, I note the absence of information that I had expected to appear in brief in the body of the manuscript, and in more detail in supplementary information. The description of 'Expert Consultation' documents the number of people consulted and notes in-person, phone, and online meetings. It also notes parenthetically that team members 'checked' for consistent application. Supplementary Note 2 merely lists the dates and places of meetings and a list of participant names.

Group deliberations may be run and facilitated in many different ways. Participants may be asked to make initial assessments independently (to avoid anchoring), followed by sharing judgements and private information, followed by revision of estimates of parameters. Alternatively, groups may be sent preliminary or previous assessments and be asked to update. Experts may or may not be provided with feedback, and be given the chance to comment on the outcomes of deliberations. The identity of experts may be anonymous for preliminary assessments (to avoid halo and dominance effects) and during deliberations, if deliberations are held remotely. Background information may or may not be shared, before or after initial judgements are made. The methods for resolving disparate estimates and for aggregating judgements make a difference to outcomes and should be documented. Uncertainty in judgements may (or may not) be elicited, recorded and reported.

I have personal preferences regarding how these things are best conducted. However, it seems important to at least document how the various elements of expert consultation were approached, and what the thinking was that went into supporting the methods used, especially against the background of recent advances in the fields of expert judgement and group deliberations.

These comments are about what is missing rather than about what was done. I would expect that these issues could be documented and the choices supported with careful reflection and documentation. I would hope to see a thoughtful assessment of the strengths and weaknesses of the tactics and procedures used here, ideally with recommendations regarding what could be done differently and potentially, better, in the next iteration.

Referee #2 (Remarks to the Author):

Overall, as I would have expected given the vast amount of effort and input that has gone into the preparation of this work, it is great, and it certainly is important and needs to be published. The key results, that amphibians are highly threatened and that by and large the threats to them are increasing, are sadly not a surprise. The data are certainly valid and the treatment of them is robust and generally well presented, though see comments to authors on some small aspects of the presentation. It is very well written and with a very few exceptions is easily understandable and has avoided excessive use of technical terminology or jargon. I did have some comments and queries, all of them really quite minor.

222/223 “amphibians are the second most threatened (40.7%) comprehensively assessed group” versus

240 “confirming amphibians as the most threatened vertebrate class”

I suppose that this means that some other group of organisms at some level has been assessed and is in the aggregate more threatened, while of classes of vertebrates, at least those that have been comprehensively assessed, amphibians are the most threatened, but I found these two statements in close proximity to be confusing and would suggest changing the wording somehow. (Added later-- It is possible to figure this out from info presented later in the paper, but one shouldn't have to.)

252 seems that it would be best to say that the 2004 assessment demonstrated that amphibians “were” the most threatened class?

266-67 “However, Red List category changes only due to new information (i.e., non-genuine changes) can introduce biases in the data.” I found this wording confusing and would suggest rewording to something like “This improved information can cause changes in the Red List categories of species that have not actually changed in status (i.e. non-genuine changes) and can thus bias the data.”

287 the wording seems odd to me, saying extinctions “by” 1980, then “in” 2004 and “in” 2022 seems to imply that the actual last individuals died in 2004 and 2022 which seems unlikely. I'm sure what is meant is that 23 species were considered extinct by 1980, with 10 more added in 2004 and 4 added in 2022. Trivial but maybe reword.

308 I suspect the semicolon after 1980 should be a comma

318-319 and other places in this section. It really seems very likely that the decrease in the negative slope of the RLI lines for most regions in the 2004+ period reflects the fact that the initial major outbreaks of global strain chytridiomycosis have had their effects on amphibian diversity, and it seems telling that the slope for the Indomalayan region, which is most coincident with the probable region of origin of the global pandemic strain as I understand it, is the only one that isn't negative during the period of the assessments. It is also possibly significant that New Guinea thus far seems to be serving as a chytridiomycosis free refuge, as documented in recent publications (DS Bower, et al. 2019. *Frontiers in Ecology and the Environment* 17 (6): 348-354) so that there is a reasonable

possibility that there will be a period of outbreak/decline in the future.

371 because the events referred to (species becoming heavily affected by disease) have occurred in the past, it seems to me that at least some of this should be in past tense, maybe “Species moving into the highest extinction risk categories have been much more likely to be...” or maybe more grammatically “Species moving into the highest extinction risk categories are much more likely to have been impacted by...”

Also, when I look at the legend for Figure 4, I think it could be a little bit clearer that the bars represent only new movements into each category during the time periods shown, or at least I think that is what they show. The present wording “Number of amphibians with deteriorating status by primary driver and Red List category at the end of each time period” might suggest to someone just skimming, anyway, that there are far fewer CR(PE) amphibians in 2022 than in 2004, if one takes deteriorating status to mean doing badly, rather than doing worse than in the previous period. Maybe (assuming my interpretation is correct) “Increases in numbers of amphibian species in each IUCN Red List category by primary driver during each time period”?

Ross A. Alford

Referee #3 (Remarks to the Author):

This is an important paper and a critical update of amphibian species' status world wide. It should have broad readership.

My main comment is that the discussion section lacks any insights based on the new data and could have been written more or less without the dataset. Please add more insights as to the broader scope and implications of the new data set.

An additional comment is that I'm not sure how the authors arrived at a 40.7% of amphibians that are threatened. $2872 \text{ species out of } 8461 = 33.9\%$ and if you exclude data deficient species (909) then $2872/7552 = 38.02\%$. Please explain these discrepancies.

Also more explanation is needed as to how "disease" caused declines prior to 1998, when the chytrid fungus (Bd) was described. I think the declines were consistent with disease, but not really proven that they were disease, except perhaps in central America and the Australian tablelands.

Minor comments:

The summary paragraph should include one sentence of introduction (not 3) and then get right into the amphibian status update (please shorten lines 230-234 to one sentence)

Line 243 - should read IUCN status deteriorations

Lines 244 and throughout - deteriorations is used throughout but without a qualifier. Should either read as declines or IUCN status deteriorations. Amphibian populations/ species do not "deteriorate"

themselves.

Line 255 - IUCN red list status

Line 258 - what are the alternatives to Bd (e.g., ranaviruses, other possibilities)

Line 266 - amphibian systematics has not "evolved" -please rephrase - tools in amphibian systematics have been improved dramatically

Line 271 - underpins is not correctly used here, should be "informs" or a similar word

Line 277 - delete "however" since the sentence is not quite an extension of the previous

Line 331-333 please reconcile/ explain why direct developers are considered at high risk for Bd, when most direct developers hatch on land and Bd is largely stream associated

Line 340 - I think the authors mean Gymniophona are poorly studied rather than poorly known (species have at least been described)

Line 343 -IUCN status

Line 343-344 - How many species?

Line 367 - IUCN status rather than "status deteriorations"

Line 369 - declined further rather than deteriorated

Line 372 - cite this statement

Line 387 - please change salamanders to salamander species

Referee #4 (Remarks to the Author):

This is an overall presentation of the results of the third Global Amphibian Assessment; data is as powerful as always due to IUCN standard methodology, and as such it also presents the widely known limitations that these assessments necessarily have – and which you also acknowledge. My congratulations for taking this out.

I only have a handful of comments that I hope will help you improve your manuscript:

- Data on breeding strategy is a bit sparse; the criteria you used for inferring them in species with no data is relatively fair, but there are reported differences between species of the same genus (I know at least one group of researcher who's assessing this). So it would be interesting to report how many species were inferred for each reproduction type, and eventually add some analyses in the Extended Data showing the patterns without these species.

- The “Red List Index” and “Genuine changes in status” sections read a bit tedious and would benefit from some small sharpening.

- I would move Extended Data Fig. 1 to the main text. Its result is as interesting as those depicted in the other figures.

- Figures 2, 3 and 4 (and Extended Data Fig 1) present colour combinations that may be difficult to perceive for colour blind people. Please check the suitability of these palettes and change them if necessary.

- Besides that, figure 2 is particularly confusing. Please avoid using the same colour, symbol and line codes in all four plates. This will make them easier to understand.

Finally, some minor points:

Line 282. The right name for the biome is “Atlantic Forest”

L287 I would say “Documented amphibian extinctions” and “continue to increase”; here note that “rise” can be understood as increasing rates, which is seemingly not the case; numbers are increasing, but luckily rates are not (unfortunately they do not seem to diminish either, though)

L619-20. This is a fair decision, but please indicate how many populations do these species have: only one, up to three, up to five, some distance threshold...

Section 2.5 Indicate if possible the sources (maps or papers) used to identify biogeographical realms and biodiversity hotspots; you know there may be some disagreements about borders between classifications.

Author Rebuttals to Initial Comments:

Point-by-point responses to referees' comments

Manuscript #2023-01-00851: Status of the world's amphibians: emerging threats and ongoing declines

We thank the editor and referees for the time they dedicated to considering our manuscript. The input we received was useful in making the following improvements:

- Improving the clarity of some sentences in the main text and one of the figure captions;
- Adding some details to the methods;
- Clarifying that the discussion refers to the findings of the study; and
- Moving the Extended Data Figure 1 to the main text.

Detailed responses to each of the referees' comments are provided below.

Referee #1 (Remarks to the Author)

This manuscript presents the results of a very considerable body of work, presenting data and analyses of international significance for amphibian ecology and the mitigation of biodiversity loss. The analysis rests on the data and judgements of a large number of professionals and the well-established IUCN Red List protocols and it has used them in a robust and conventional way.

However, I note the absence of information that I had expected to appear in brief in the body of the manuscript, and in more detail in supplementary information. The description of 'Expert Consultation' documents the number of people consulted and notes in-person, phone, and online meetings. It also notes parenthetically that team members 'checked' for consistent application. Supplementary Note 2 merely lists the dates and places of meetings and a list of participant names.

Group deliberations may be run and facilitated in many different ways. Participants may be asked to make initial assessments independently (to avoid anchoring), followed by sharing judgements and private information, followed by revision of estimates of parameters. Alternatively, groups may be sent preliminary or previous assessments and be asked to update. Experts may or may not be provided with feedback, and be given the chance to comment on the outcomes of deliberations. The identity of experts may be anonymous for preliminary assessments (to avoid halo and dominance effects) and during deliberations, if deliberations are held remotely. Background information may or may not be shared, before or after initial judgements are made. The methods for resolving disparate estimates and for aggregating judgements make a difference to outcomes and should be documented. Uncertainty in judgements may (or may not) be elicited, recorded and reported.

I have personal preferences regarding how these things are best conducted. However, it seems important to at least document how the various elements of expert consultation were approached, and what the thinking was that went into supporting the methods used, especially against the background of recent advances in the fields of expert judgement and group deliberations.

These comments are about what is missing rather than about what was done. I would expect that these issues could be documented and the choices supported with careful reflection and documentation. I would hope to see a thoughtful assessment of the strengths and weaknesses of the tactics and procedures used here, ideally with recommendations regarding what could be done differently and potentially, better, in the next iteration.

Response:

We agree that more detail is needed on the elicitation methods used in the GAA2. While we did not use formal elicitation methods, we have added several sentences to section 1.2. Expert Consultation (line 566) to better explain the data with which contributing experts were presented, how we reduced expert bias and prevented our reliance on a particular person's contributions, and captured uncertainty in the data. In addition, we followed the IUCN Red List Guidelines on Data Quality in using a standardized approach in recording data uncertainty, data qualifiers, and differences in risk tolerance between contributing experts. The Guidelines chapter are now referenced in this section as well.

Further, section 1.3 Assessment Finalising (line 571) reports our efforts to ensure consistent data gathering and application of the IUCN categories and criteria between species and between countries.

Due to the time required to address the growing number of species that require assessment and reassessment in each GAA cycle, and the number of decisions required by experts per data field in each assessment (e.g. geographic range, population size and trends, impact of threats, etc.), it is not practical to use formal elicitation methods at a global scale. Moreover, the majority of the contributing experts volunteer their time (ie. it is not in their job descriptions to participate nor do they receive funds in exchange for their time) and we must work within a limited timeframe. Expert willingness to participate and their availability are aspects that we must balance in addition to producing high quality assessments.

We are aware of a small subset of IUCN Red List assessments that have used a modified Delphi technique (e.g. Horseshoe crab assessments). However these tend to be much less speciose groups compared to amphibians, and therefore a more feasible task.

Referee #2 (Remarks to the Author)

Overall, as I would have expected given the vast amount of effort and input that has gone into the preparation of this work, it is great, and it certainly is important and needs to be published. The key results, that amphibians are highly threatened and that by and large the threats to them are increasing, are sadly not a surprise. The data are certainly valid and the treatment of them is robust and generally well presented, though see comments to authors on some small aspects of the presentation. It is very well written and with a very few exceptions is easily understandable and has avoided excessive use of technical terminology or jargon. I did have some comments and queries, all of them really quite minor.

- 222/223 “amphibians are the second most threatened (40.7%) comprehensively assessed group” versus 240 “confirming amphibians as the most threatened vertebrate class”. I suppose that this means that some other group of organisms at some level has been assessed and is in the aggregate more threatened, while of classes of vertebrates, at least those that have been comprehensively assessed, amphibians are the most threatened, but I found these two statements in close proximity to be confusing and would suggest changing the wording somehow. (Added later--It is possible to figure this out from info presented later in the paper, but one shouldn't have to.)

Response: We agree this wording is confusing and have reworded this section to make it more clear.

- 252 seems that it would be best to say that the 2004 assessment demonstrated that amphibians “were” the most threatened class?

Response: We agree and have changed the verb tense in the manuscript. We also added text to line 284 to reinforce that GAA2 confirms amphibians remain the most threatened class of vertebrates.

- 266-67 “However, Red List category changes only due to new information (i.e., non-genuine changes) can introduce biases in the data.” I found this wording confusing and would suggest rewording to something like “This improved information can cause changes in the Red List categories of species that have not actually changed in status (i.e. non-genuine changes) and can thus bias the data.”

Response: We agree that this was confusing and have made changes to the text incorporating the reviewer's suggestion.

- 287 the wording seems odd to me, saying extinctions “by” 1980, then “in” 2004 and “in” 2022 seems to imply that the actual last individuals died in 2004 and 2022 which seems unlikely. I'm sure what is meant is that 23 species were considered extinct by 1980, with 10 more added in 2004 and 4 added in 2022. Trivial but maybe reword.

Response: We agree that the way the text was written could lead to confusion in the number of extinctions and made changes to the text addressing the reviewer's suggestion.

- 308 I suspect the semicolon after 1980 should be a comma

Response: We cannot see this semicolon. Perhaps this was an issue introduced by the conversion of a Word document to Adobe PDF? In any case, we confirm that “1980” on line 308 is followed by a comma.

- 318-319 and other places in this section. It really seems very likely that the decrease in the negative slope of the RLI lines for most regions in the 2004+ period reflects the fact that the initial major outbreaks of global strain chytridiomycosis have had their effects on amphibian diversity, and it seems telling that the slope for the Indomalayan region, which is most coincident with the probable region of origin of the global pandemic strain as I understand it, is the only one that isn't negative during the period of the assessments. It is also possibly significant that New Guinea thus far seems to be serving as a chytridiomycosis free refuge, as documented in recent publications (DS Bower, et al. 2019. *Frontiers in Ecology and the Environment* 17 (6): 348-354) so that there is a reasonable possibility that there will be a period of outbreak/decline in the future.

Response: We agree with the reviewer's reflections on the results presented in lines 318-319. The reviewer's point about potential future disease outbreaks in New Guinea has been incorporated.

- 371 because the events referred to (species becoming heavily affected by disease) have occurred in the past, it seems to me that at least some of this should be in past tense, maybe “Species moving into the highest extinction risk categories have been much more likely to be...” or maybe more grammatically “Species moving into the highest extinction risk categories are much more likely to have been impacted by...”

Response: The reviewer's suggested change has been incorporated into the manuscript.

- Also, when I look at the legend for Figure 4, I think it could be a little bit clearer that the bars represent only new movements into each category during the time periods shown, or at least I think that is what they show. The present wording “Number of amphibians with deteriorating status by primary driver and Red List category at the end of each time period” might suggest to someone just skimming, anyway, that there are far fewer CR(PE) amphibians in 2022 than in 2004, if one takes deteriorating status to mean doing badly, rather than doing worse than in the previous period. Maybe (assuming my interpretation is correct) “Increases in numbers of amphibian species in each IUCN Red List category by primary driver during each time period”?

Response: The reviewer is generally correct in their understanding of the figure. Edits have been made to the caption to more clearly explain what it represents.

Ross A. Alford

Referee #3 (Remarks to the Author)

- This is an important paper and a critical update of amphibian species' status world wide. It should have broad readership.
- My main comment is that the discussion section lacks any insights based on the new data and could have been written more or less without the dataset. Please add more insights as to the broader scope and implications of the new data set.

Response: The discussion presents recommendations in response to the GAA2 data analyses presented in the manuscript and the collective knowledge and expertise of the co-authors was drawn upon to inform the discussion. However, this could have been better stated. We have made substantive changes to highlight more clearly what are the results of our study, rather than those supported by other works.

- An additional comment is that I'm not sure how the authors arrived at a 40.7% of amphibians that are threatened. 2872 species out of 8461 = 33.9% and if you exclude data deficient species (909) then $2872/7552 = 38.02\%$. Please explain these discrepancies.

Response: The method for calculating the percentage of threatened species is explained in section 4.1 of the Methods. It is noted that this was not clear in the manuscript and this has now been added. Indeed the reviewer is correct, DD species are removed from the total number of species, as are EX species. The number of EW species (2 in this case) are added to the number of threatened species. This gives $2875/7065 = 0.4067$ or 40.7%. The confusion has arisen because the reviewer has calculated the proportion relative to the number of described species at the time the manuscript was written, rather than the number assessed in the GAA2. These numbers differ due to the rapid pace of new species descriptions. When the manuscript was written, 8,461 species were known to science today, there are 8,615 species. We have revised both the number of species and the percentage covered by the GAA2 to reflect this.

- Also more explanation is needed as to how "disease" caused declines prior to 1998, when the chytrid fungus (*Bd*) was described. I think the declines were consistent with disease, but not really proven that they were disease, except perhaps in central America and the Australian tablelands.

Response: This study brings together the knowledge and published data of a large number of experts worldwide to assess extinction risk. Many studies (e.g., see references 2,3,4,5 and 6) now link amphibian population declines prior to the description of *Bd* with chytridiomycosis based on tested museum specimens, or when the timing and patterns of decline and disappearances are consistent with other species confirmed with the disease, and our assessments draw upon the results of these studies.

Minor comments:

- The summary paragraph should include one sentence of introduction (not 3) and then get right into the amphibian status update (please shorten lines 230-234 to one sentence)

Response: These sentences have been significantly shortened and combined into one to address this concern.

- Line 243 - should read IUCN status deteriorations

Response: We have altered the text to read “major driver of increases in extinction risk” for clarity.

- Lines 244 and throughout - deteriorations is used throughout but without a qualifier. Should either read as declines or IUCN status deteriorations. Amphibian populations/ species do not "deteriorate" themselves.

Response: The term deterioration is used specifically in this manuscript in reference to IUCN Red List categories, which indicate a species relative extinction risk. However, we accept the reviewer’s input regarding potential confusion and have reworded line 244 to avoid giving readers the impression we are speaking of population deterioration or declines.

Further, we were very careful to use official IUCN terms throughout the manuscript. The IUCN does not recognize the term “IUCN status”. Thus, we cannot fully apply the reviewer’s recommendation. Instead, the reviewer and editor will see that we have consistently used terms such as “status deteriorations”, “deteriorating status”, “improvements in status” and “IUCN Red List”, “Red List category”, and “category”. We do not use the term “declining status” for consistency and to avoid confusion with population declines, which are a separate concept and data field.

- Line 255 - IUCN red list status

Response: Please see response to comment for line 244.

- Line 258 - what are the alternatives to Bd (e.g., ranaviruses, other possibilities)

Response: The language on this line was carefully crafted by the co-authors to reflect that, while there is ample evidence that Bd was implicated in many of these enigmatic declines, not all of the enigmatic declines have been confirmed as caused by Bd. However, because the nature and timing of declines with unconfirmed causes correlate and coincide with those caused by Bd, it is highly likely that Bd was implicated. Additionally, some studies suggest that other pressures, threats, and changes (e.g. climate change) in species’ habitats could have exacerbated the effect of and/or coincided with Bd’s arrival, thus this cannot be ruled out. We hope that the current wording reflects this data uncertainty.

- Line 266 - amphibian systematics has not "evolved" -please rephrase - tools in amphibian systematics have been improved dramatically

Response: We changed the word from “evolved” to “progressed”

- Line 271 - underpins is not correctly used here, should be "informs" or a similar word

Response: We agree with the reviewer and have altered the text in line with this suggestion.

- Line 277 - delete "however" since the sentence is not quite an extension of the previous

Response: We agree and have made the reviewer’s suggested edit.

- Line 331-333 please reconcile/ explain why direct developers are considered at high risk for Bd, when most direct developers hatch on land and Bd is largely stream associated

Response: The reviewer is correct in citing the paragraph’s report of extinction risk being highest for direct developers. However, it does not give any reason for this high risk in direct developers. Instead, it explains that the slowing of the larval developer RLI during 2004-2022 is related to the timing of Bd infections. The reason for direct developer’s high risk is discussed on lines 382-385, which read “Decreased rainfall due to climate change in the Wet Tropics of Australia and Brazil’s Atlantic Forest is also predicted to reduce the reproductive success of direct-developing frogs (e.g., in the genera *Cophixalus* and *Brachycephalus*) because of their dependence on high levels of soil and leaf-litter moisture to prevent egg desiccation.”

Additional text has been added to line 333 explaining that additional information is needed to interpret these trends.

- Line 340 - I think the authors mean Gymniophona are poorly studied rather than poorly known (species have at least been described)

Response: We agree that this is a useful distinction to make, which better reflects the intended meaning of the sentence. We have made the suggested change to the text.

- Line 343 -IUCN status

Response: Please see response to comment for line 244

- Line 343-344 - How many species?

Response: The number of species that deteriorated in status are detailed in the last sentence of this paragraph and then status improvements are detailed in a later paragraph. Because some species have changed category (either up or down) across both time periods, it is potentially confusing to quote a total number of species that have changed category. Thus, this is considered the most accurate way to present the number of species rather than a one off figure at the beginning.

- Line 367 - IUCN status rather than "status deteriorations"

Response: Please see our response to comment for line 244.

- Line 369 - declined further rather than deteriorated

Response: This sentence refers specifically to results related to deteriorations in status, not population declines. As such, “deteriorated” is the correct term here. See also comment for line 244 where the specific terms are explained.

- Line 372 - cite this statement

Response: The first half of this statement is a result from the GAA2 data analysis and Fig. 4 is now cited. The second half of the sentence is now also cited.

- Line 387 - please change salamanders to salamander species

Response: The suggested change has been made to the manuscript.

Referee #4 (Remarks to the Author)

This is an overall presentation of the results of the third Global Amphibian Assessment; data is as powerful as always due to IUCN standard methodology, and as such it also presents the widely known limitations that these assessments necessarily have – and which you also acknowledge. My congratulations for taking this out.

I only have a handful of comments that I hope will help you improve your manuscript:

- Data on breeding strategy is a bit sparse; the criteria you used for inferring them in species with no data is relatively fair, but there are reported differences between species of the same genus (I know at least one group of researcher who’s assessing this). So it would be interesting to report how many species were inferred for each reproduction type, and eventually add some analyses in the Extended Data showing the patterns without these species.

Response: We agree on both points that a) it is correct that there are differences in breeding strategy between species of the same genus, and b) it would be an interesting result to report. However, whether or not the breeding strategy has been inferred was captured in a descriptive text field and it would require us to go through all 8,011 assessments and recode them in order to determine this information, a very time-consuming process. Fortunately, where the breeding strategy is known to differ from congeners, this is recorded within both the descriptive text field and the categorical fields in our database. It is only for the subset of species for which the breeding strategy has not been observed or confirmed that the strategy is inferred. When breeding strategies are confirmed, this information will be incorporated into re-assessments under future GAA initiatives.

- The “Red List Index” and “Genuine changes in status” sections read a bit tedious and would benefit from some small sharpening.

Response: We appreciate this feedback and recognize that both of the sections in question are packed full of results and references to figures, which can make the text dry at times. We are pleased to report that the co-authors spent a considerable amount of time honing the text prior to submission to ensure that critical results and concepts were clearly communicated. An additional consideration is that English is not the first language of the majority of our co-authors. Thus, their feedback was crucial to crafting language that could easily be understood by the audiences of Nature for whom this is also the case. In addition, textual changes have been incorporated in response to the comments from Reviewers 1, 2, and 3. We are confident that the final layout of the paper will help to visually break up result-dense sections of text and illustrate the results in a compelling manner.

- I would move Extended Dat Fig. 1 to the main text. Its result as interesting as those depicted in the other figures.

Response: We wholeheartedly agree with the reviewer on the importance of the results in Extended Data Figure 1, as well as the interest it would generate among the readers. Prior to submission, several draft versions of the manuscript included Extended Data Fig. 1 in the main text. However, following co-author feedback, we replaced it with Figure 1, which is also very compelling and will be of high interest to our audience. We regret that space constraints likely prevent us from moving Extended Data Fig. 1 to the main text. However, should the final layout allow for it, we would be glad to do so.

- Figures 2, 3 and 4 (and Extended Data Fig 1) present colour combinations that may be difficult to perceive for colour blind people. Please check the suitability of these palettes and change them if necessary.

Response: We ran it through a colour blindness simulator and it informed us that the exact shades of colours used in the figure were distinct enough for colour blind readers to be able to distinguish among them.

- Besides that, figure 2 is particularly confusing. Please avoid using the same colour, symbol and line codes in all four plates. This will make them easier to understand.

Response: We have changed the symbols and some of the colours such that they are now all different in each graph and hopefully they are now more easily distinguished.

Finally, some minor points:

- Line 282. The right name for the biome is “Atlantic Forest”

Response: This is a good catch. We have edited the text in line with the reviewer’s suggestion.

- L287 I would say “Documented amphibian extinctions” and “continue to increase”; here note that “rise” can be understood as increasing rates, which is seemingly not the case; numbers are increasing, but luckily rates are not (unfortunately they do not seem to diminish either, though)

Response: We agree with the reviewer’s suggestions and have incorporated them into the text.

- L619-20. This is a fair decision, but please indicate how many populations do these species have: only one, up to three, up to five, some distance threshold...

Response: The 53 species without a distribution map were assessed in the Data Deficient (DD) IUCN Red List category. While they do not have a map, any available information about the possible distribution and/or population of these species is documented in their IUCN Red List assessments. However, DD species are defined by a significant lack of data such that even the number of populations is unknown, let alone the population size or provenance of the records from which the species was described.

- Section 2.5 Indicate if possible the sources (maps or papers) used to identify biogeographical realms and biodiversity hotspots; you know there may be some disagreements about borders between classifications.

Response: This is an excellent point. The sources have been added to this section and the relevant reference list.

Reviewer Reports on the First Revision:

Referees' comments:

Referee #1 (Remarks to the Author):

The paper is improved, but the authors have missed the point of some of my comments. I apologize for the lack of clarity. I have attempted to provide a clearer set of comments in the attached document.

In short I am seeking a fuller explanation of what was done, to the point that the work could be replicated, together with some reflection on what may be done better next time. I hope the authors are able to take this on board, to enhance their very significant achievement.

Referee #2 (Remarks to the Author):

The authors have responded well to my comments and those of other reviewers on the first draft I sighted, and I have no remaining issues to comment on.

The results are important and highly significant, presented well, and appropriately analysed. The conclusions are robust and the references are appropriate.

I am satisfied that the paper as presently constituted is an important contribution, and congratulate the authors on a major piece of work.

Ross A. Alford

Referee #3 (Remarks to the Author):

This is a highly significant piece of work - not that it is particularly novel research, but rather it provides a key update on the status of the worlds amphibians. The revised version is acceptable for publication in my view since the authors responded well to reviewer comments.

Referee #4 (Remarks to the Author):

Your manuscript has improved since last version, and I commend you for that. I'm happy that you finally decided to include the figure of the number of species affected by each threat into the main text (although you say the opposite in the comments to the reviewers).

However, I disagree with you in certain parts of the rebuttal, which I indicate below. These do not affect the overall quality of the work, although I still believe that they would improve the readability and impact of the manuscript.

- I'm not a native English speaker myself, and I still find the two sections I mentioned before (RLI and Genuine changes) a bit tedious. However, there is little you can do if you want to be descriptive to the level you do, and I see the value of mentioning certain things in the main text of a spotlight paper such as this one. The current version of the text reads well, which is the most important thing.

- I see that extracting numbers for the species for which breeding strategy was estimated would be difficult. But it is not impossible, it would mean reading a bit more than 8K assessment, which divided between more than 120 authors leaves about 70 assessments each. That seems doable to me, and fair to do it for a revision if you truly think this is an important point, as you say in the rebuttal letter.

- Same as above, reading the 53 assessments of DD species without map to qualify them as having 1, 2, 3, or no known populations in the wild does not seem difficult to do. Many databases, such as BIEN, have minimum thresholds to present maps just as known presence points. I concur that in DD it is likely that only one population (and often only one individual) is known, but the text should indicate that, or at least say "when no reliable information on their distribution is available, or no populations are known besides the one out of which the species was described."

Author Rebuttals to First Revision:

Point-by-point responses to referees' comments

Manuscript #2023-01-00851B: Ongoing declines for the world's amphibians in the face of emerging threats

Detailed responses to each of the referees' comments are provided below.

Referee #1 (Remarks to the Author):

The paper is improved, but the authors have missed the point of some of my comments. I apologize for the lack of clarity. I have attempted to provide a clearer set of comments in the attached document.

In short I am seeking a fuller explanation of what was done, to the point that the work could be replicated, together with some reflection on what may be done better next time. I hope the authors are able to take this on board, to enhance their very significant achievement.

[Text from the email attachment:]

The point-by-point response seems to have overlooked several points. I'll list them separately so that they are clearer.

- Were participants asked to make initial assessments independently of other experts, or not?
- Did participants discuss their estimates, or not? If so, were discussions before or after they made an initial judgement?
- Alternatively, did facilitators discuss each case (taxon) in an unstructured fashion and reach a behavioral consensus (i.e., group agreement on the outcome based on face to-face discussion, sometimes termed forced consensus)?
- Were experts provided with feedback about the assessments?
- Were experts given the opportunity to revise their initial estimates after being provided feedback?
- Were experts given the chance to comment on the outcomes of deliberations?
- Were the initial judgements of experts anonymous (not the case if behavioral consensus was used)?
- Was background information on each taxon shared with experts prior to them making judgements?
- Were experts aware of prior IUCN classifications of the taxa (probably yes, but please clarify)?
- How were disparate estimates resolved or aggregated?
- Uncertainty in judgements was elicited and recorded. What was done with this

information?

- Did the uncertainties encompass differences of opinion between experts?

Authors' comment

Due to the time required to address the growing number of species that require assessment and reassessment in each GAA cycle, and the number of decisions required by experts per data field in each assessment (e.g, geographic range, population size and trends, impact of threats, etc.), it is not practical to use formal elicitation methods at a global scale. Moreover, the majority of the contributing experts volunteer their time (ie. it is not in their job descriptions to participate nor do they receive funds in exchange for their time) and we must work within a limited timeframe. Expert willingness to participate and their availability are aspects that we must balance in addition to producing high quality assessments.

Referee response

This comment raises two points.

1. I admire the breadth and scope of this undertaking and the voluntary contributions of the many participants. I fully understand the need to be efficient and get the job done. The issue is not that, but rather, it is that many of the individual questions listed above refer to facilitation steps that could be implemented without any additional work. In fact, several of them reduce workload and enhance participants' trust in the outcomes. My reaction to the comment is that it may reflect a lack of familiarity with these tools. There's no need to be defensive about what was done. It was a Herculean task. Rather, I would like to see some reflection on what could have been done better, so that the approach is improved at the next iteration.
2. In the spirit of good scientific method, the description of what was done should be sufficiently detailed that it could be replicated in full. In its current form, I would struggle to know exactly how things were conducted. Dealing with the things arising in the question above in the text of the Methods section would help.

Authors' response: The provision of additional clarity from the referee is appreciated. In response to point 1 above, we have added the following new paragraph to Methods section 1.2 addressing desirable improvements for future Global Amphibian Assessments:

“We acknowledge that more formal elicitation methods, such as structured expert elicitation, can identify and reduce potential sources of bias and error among experts when contributing data and

making judgements. This structured process could prove valuable for future IUCN Red List assessment processes, particularly for high profile or contentious taxa, though it may be impractical for less contentious taxa due to the amount of time required⁴².”

42. McBride, M.F., Garnett, S.T., Szabo, J.K., Burbidge, A.H., Butchart, S.H., Christidis, L., Dutson, G., Ford, H.A., Loyn, R.H., Watson, D.M. and Burgman, M.A., 2012. Structured elicitation of expert judgments for threatened species assessment: a case study on a continental scale using email. *Methods in Ecology and Evolution*, 3(5), pp.906-920.

In response to point 2 above, we have added more detail of the expert consultation stage in view of addressing the twelve questions provided in the referee’s attachment (pasted above in this document). However, it must be noted that, because formal elicitation methods are not commonly used in the practice of red listing, the additional information added to the Methods cannot go into the detail requested in the referee’s twelve questions.

–

Authors’ comment

A considerable amount of effort went into engaging with a diversity of experts across several axes (e.g., gender, age, geography, type of expertise) so as to reach the widest range of experts as possible and minimise reliance on any individual expert.

Referee response

Wonderful. Please give details (a table with a breakdown of ages, genders, geography, and expertise would be ideal).

Authors’ response:

Presenting the demographic information of participants would suggest that these are dependent variables in our study, which they are not. For example, we bring younger and older participants to our workshops for various reasons. Younger experts often have more recent field data, while older participants have more historical knowledge of the species and localities. In addition, the workshop provides an opportunity for young experts to connect and learn from older experts. We see these settings as a way to build future capacity in conservation. The lead authors believe that collection and publication of this information would mislead a reader to believe that these demographics affected the outcomes of the study, which we do not agree with. Many other publications have done similar studies of other taxonomic groups and this data has never been published. We would also be concerned that participating experts would find this an unnecessary intrusion of their privacy, and suggest that their contributions are less about scientific data and more about speculation.

The following paragraph was added to the Methods to capture our reflection on improvements that could be made in future:

“Future Global Amphibian Assessment initiatives would benefit from increasing the breadth of expertise engaged. In particular, increased participation from conservation organisations and natural resource management or wildlife branches of governments should be targeted. Participants of both the first and second Global Amphibian Assessment were often members of academic institutions with expertise on herpetology, biogeography, taxonomy, etc, as they were often the only scientists to have ever seen the species and visited known sites, and because they were typically experts in the species of the region or family of species being assessed. That said, participants without expertise in herpetology but with relevant expertise on regional threatening processes such as climate projections and wildlife trade, conservation planning, policy, and implementation have the potential to improve the quality of the threat and conservation fields in the assessments.”

–

Authors’ comment

When uncertainty in the data and differences in risk tolerance between contributing experts was encountered, this was captured in the assessment as a range of possible values or scenarios.

Referee response

Again, wonderful. This suggests that answers to some of the questions above are straight forward. What was done with this information. For instance, I see an expression at line 778. Should it play a role there?

Authors’ response:

The following paragraph has been added to the Methods section 1.2:

“Uncertainty in the data and differences in risk tolerance between contributing experts were documented as a range of values in accordance with Section 3.2.5 of the IUCN Red List Guidelines. When this resulted in a range of possible Red List categories being met (e.g., Endangered–Critically Endangered), the range of categories was captured in the assessment rationale and a single category was chosen with clear justification for the decision, including whether an evidentiary or precautionary attitude was adopted. Where the uncertainty was deemed to be too great, the category of Data Deficient was applied in compliance with Section 10.3 of the IUCN Red List Guidelines.”

–

Authors’ comment

Red List Guidelines. In many cases where direct observational data were not available, data fields (e.g., population size and severity of threats) were derived through expert estimation or inference using basic facilitation techniques.

Referee response

Relevant to the comments above, this description is incomplete. I’m guessing that ‘basic facilitation’ refers to unstructured discussion and behavioral consensus. Maybe not. As noted above, please provide full details so that the process could be repeated. And please reflect on whether the process could be made more transparent, repeatable, and better trusted if some of the steps alluded to above were included in future iterations.

Authors’ response:

Apologies for the confusion, but when referring to ‘basic facilitation’, we do not mean no facilitation. Discussions are guided by IUCN trained facilitators and they do not rely on behavioral consensus. Instead, extinction risk is determined by application of the IUCN Red List Categories and Criteria to the data provided. The following paragraph was added to the Methods section 1.2 should make clear what facilitation was provided by highly qualified staff:

“The expert consultation process was led by IUCN Red List trained facilitators and followed the IUCN Rules of Procedure⁴¹: 1) expert validation of the data in the draft assessments, 2) contribution of missing data and/or revision of data with suitable justification, and 3) group discussion and application of the IUCN Red List Categories and Criteria to the data.”

41. IUCN. Rules of Procedure for IUCN Red List Assessments 2017–2020. Version 3.0. Approved by the IUCN SSC Steering Committee in September 2016. http://cmsdocs.s3.amazonaws.com/keydocuments/Rules_of_Procedure_for_Red_List_2017-2020.pdf (2016).

Authors’ comment

The supporting data and Red List categories were finalised by an ARLA team member who also performed checks to ensure the IUCN categories and criteria were applied in a consistent manner to the species within a particular region, but also between ASG regions. Consistency was also sought for species with similar traits or co-occurring species.

Referee response

Also wonderful, but in pursuit of clarity and repeatability, can you say in a little more detail what ‘consistency’ means and what an ‘inconsistent’ result looked like?

Authors’ response:

Consistency and inconsistency refer to the application of the IUCN categories and criteria to the data provided by experts. A consistent result would come from the application of the categories and criteria in the same manner to the same data, resulting in the same extinction risk assessment. An inconsistent result would be obtained if different categories were determined for two or more species with very similar data. When the ARLA detected an inconsistent result, assessments were revisited with data contributors to reconcile any discrepancies. We have applied this information to Methods section 1.3 which now reads:

“The supporting data and Red List categories were finalised by an ARLA team member who also performed checks to ensure the IUCN categories and criteria were applied in a consistent manner to the species within a particular region, but also between ASG regions. An example of an inconsistent result is when different Red List categories were determined for two or more species with very similar data. Consistency was also sought for species with similar traits or co-occurring species. If inconsistency was detected, assessments were revisited with data contributors to reconcile any discrepancies.”

Inconsistencies in the application of the categories and criteria most commonly occurred when experts consider that they have very little data for a species and are, thus, likely to suggest that the species is Data Deficient. For example, in well researched regions such as the USA and Australia, the most poorly known species are often suggested by experts as being Data Deficient as they have far less data compared to comprehensively studied species. Yet, when compared to a region where there has been considerably less research, a species with that same amount of data may be considered well-known and experts may feel comfortable assessing it. Our trained facilitators led workshops all over the world and were aware of these inconsistencies. Hence, when collating data from regions, these facilitators were best placed to check for such inconsistencies in application of the categories and criteria.

Referee #2 (Remarks to the Author):

The authors have responded well to my comments and those of other reviewers on the first draft I sighted, and I have no remaining issues to comment on.

The results are important and highly significant, presented well, and appropriately analysed. The conclusions are robust and the references are appropriate.

I am satisfied that the paper as presently constituted is an important contribution, and congratulate the authors on a major piece of work.

Ross A. Alford

Referee #3 (Remarks to the Author):

This is a highly significant piece of work - not that it is particularly novel research, but rather it provides a key update on the status of the world's amphibians. The revised version is acceptable for publication in my view since the authors responded well to reviewer comments.

Referee #4 (Remarks to the Author):

Your manuscript has improved since last version, and I commend you for that. I'm happy that you finally decided to include the figure of the number of species affected by each threat into the main text (although you say the opposite in the comments to the reviewers).

However, I disagree with you in certain parts of the rebuttal, which I indicate below. These do not affect the overall quality of the work, although I still believe that they would improve the readability and impact of the manuscript.

- I'm not a native English speaker myself, and I still find the two sections I mentioned before (RLI and Genuine changes) a bit tedious. However, there is little you can do if you want to be descriptive to the level you do, and I see the value of mentioning certain things in the main text of a spotlight paper such as this one. The current version of the text reads well, which is the most important thing.

Authors' response: We appreciate the referee's feedback on these sections and their understanding that we have taken great care to word them in such a way as to provide a detailed and accurate description of the concepts.

- I see that extracting numbers for the species for which breeding strategy was estimated would be difficult. But it is not impossible, it would mean reading a bit more than 8K assessments, which divided between more than 120 authors leaves about 70 assessments each. That seems doable to me, and fair to do it for a revision if you truly think this is an important point, as you say in the rebuttal letter.

Authors' response:

Unfortunately, this task is far more difficult than it may first appear. When a species' breeding strategy has not been directly observed, it is common and accepted practice amongst amphibian experts to assume that related species (such as those in the same genus) exhibit the same breeding strategy (although this is not the case for all genera). Considering the often cryptic and secretive nature of species, inference is often the only possibility. As much as possible, we have tried to document in the assessments when a breeding strategy is known versus inferred or suspected based on expert knowledge of similar species. However, there have undoubtedly been occasions that such inferences have not been documented and the breeding strategy has been documented as known, such that teasing out inferred breeding strategies will not be possible for all species.

- Same as above, reading the 53 assessments of DD species without map to qualify them as having 1, 2, 3, or no known populations in the wild does not seem difficult to do. Many databases, such as BIEN, have minimum thresholds to present maps just as known presence points. I concur that in DD it is likely that only one population (and often only one individual) is known, but the text should indicate that, or at least say "when no reliable information on their distribution is available, or no populations are known besides the one out of which the species was described."

Authors' response:

Each of the 53 species with no map are accompanied by a completed assessment documenting the population information for the species (if any exists) and the reasons why a map cannot be completed. In doing so, we are in compliance with the IUCN Guidelines recommended protocol for instances when a map for a species should not be included.

However, in our Methods section, we were not clear regarding just how poor the data are for these 53 species. Thus, we have now edited the sentence to read "There are 53 species in the GAA2 without distribution maps as the taxon is known only from one or more specimens with no or extremely uncertain locality information."

Reviewer Reports on the Second Revision:

Referee #1

(Remarks to the Author)

The manuscript is improved, from the perspective of the methods deployed for expert elicitation and aggregation. The additional detail is enough to provide confidence that the approach could be replicated and that the authors have made a substantial effort to anticipate and mitigate bias.

Without wanting to push things too far, there's one unfortunate claim in the authors' responses. They say, 'Due to the time required to address the growing number of species ..., it is not practical to use formal elicitation methods at a global scale. Moreover, the majority of the contributing experts volunteer their time ... and we must work within a limited timeframe. '. I am fully aware of these constraints. I merely implore the authors to contemplate the potential to improve their approach without adding ANY additional burden on the contributors. I can ample room for improvement and hope they will take up the challenge with some rethinking of the structure and organisation of their initiative.

The only remaining issue is to clarify whether any of the contributors were given an opportunity to comment or to revise their initial estimates, once they had a chance to discuss differences and to see the judgements of others. I presume not, given the time constraints, but this should be made clear - it would only require a note.